# Notch signaling functions in noncanonical juxtacrine manner in platelets to amplify thrombogenicity

**Susheel N Chaurasia, Mohammad Ekhlak, Geeta Kushwaha[†], Vipin Singh, Ram L Mallick[‡], Debabrata Dash***

Center for Advanced Research on Platelet Signaling and Thrombosis Biology, Department of Biochemistry, Institute of Medical Sciences, Banaras Hindu University, Varanasi, India

**\*For correspondence:**
ddash.biochem@gmail.com

**Present address:** [†]Department of Biochemistry, Vardhman Mahavir Medical College and Safdarjung Hospital, New Delhi, India; [‡]Department of Biochemistry, Birat Medical College and Teaching Hospital, Biratnagar, Nepal

**Competing interest:** The authors declare that no competing interests exist.

## Abstract

**Background:** Notch signaling dictates cell fate decisions in mammalian cells including megakaryocytes. Existence of functional Notch signaling in enucleate platelets remains elusive.

**Methods:** Transcripts/peptides of Notch1 and Delta-like ligand (DLL)–4 were detected in platelets isolated from human blood by RT-qPCR, Western analysis and flow cytometry. Platelet aggregation, granule secretion and platelet-leukocyte interaction were analyzed by lumi-aggregometry and flow cytometry. Platelet-derived extracellular vesicles were documented with Nanoparticle Tracking Analyzer. Platelet thrombus on immobilized collagen was quantified using microfluidics platform. Intracellular calcium was monitored by fluorescence spectrophotometry. Whole blood coagulation was studied by thromboelastography. Ferric chloride-induced mouse mesenteric arteriolar thrombosis was imaged by intravital microscopy.

**Results:** We demonstrate expression of Notch1, its ligand DLL-4 and their respective transcripts in human platelets. Synthesis and surface translocation of Notch1 and DLL-4 were upregulated by thrombin. DLL-4, in turn, instigated neighbouring platelets to switch to 'activated' phenotype through cleavage of Notch receptor and release of its intracellular domain (NICD), which was averted by inhibition of γ-secretase and phosphatidylinositol-3-kinase (PI3K). Inhibition of Notch signaling, too, restrained agonist-induced platelet activation, and significantly impaired arterial thrombosis in mice. Strikingly, prevention of DLL-4-Notch1 interaction by a blocking antibody abolished platelet aggregation and extracellular vesicle shedding induced by thrombin.

**Conclusions:** Our study presents compelling evidence in support of non-canonical juxtacrine Notch signaling within platelet aggregates that synergizes with physiological agonists to generate occlusive intramural thrombi. Thus, Notch pathway can be a potential anti-platelet/anti-thrombotic therapeutic target.

**Funding:** Research was supported by grants received by DD from JC Bose Fellowship (JCB/2017/000029), ICMR (71/4/2018-BMS/CAR), DBT (BT/PR-20645/BRB/10/1541/2016) and SERB (EMR/2015/000583). SNC, ME and VS are recipients of ICMR-Scientist-C, CSIR-SRF and UGC-SRF support, respectively. Funders had no role in design, analysis and reporting of study.

## Editor's evaluation

Advances in the discovery of novel anti-platelet therapeutics remains an unmet need. This manuscript by Chaurasia et al., describes a novel signaling pathway involving Notch1 and its ligand, Δ-like ligand-4 (DLL4) in driving platelet activation and thrombus formation. The authors provide convincing mechanistic studies to show that blockade of this pathway may serve as a new

therapeutic approach to prevent/treat thrombosis. The work will be of great interest to individuals in the hematology and thrombosis field.

## Introduction

Notch signaling, one of the evolutionarily conserved pathways in mammals, is a central regulator of cell fate determinations through cell-to-cell interactions (*Kopan and Ilagan, 2009*; *Guruharsha et al., 2012*) that critically influences cell proliferation, differentiation and apoptosis (*Miele and Osborne, 1999*). Signaling is induced through binding of five independent ligands, Delta-like ligands (DLL)–1, 3, 4 and Jagged (Jag)–1 and –2 (*Kopan and Ilagan, 2009*), to four isoforms of cognate Notch receptors, Notch1 to Notch4, on surface of adjacent cells. Binding incites sequential α- and Y-secretase-mediated proteolytic events releasing the intracellular domain of Notch receptor (NICD) that initiates downstream effects of Notch activation (*Maillard et al., 2003*; *Blanpain et al., 2006*; *Qiao and Wong, 2009*; *Andersson et al., 2011*).

Platelets are circulating blood cells having central role in hemostasis and pathological thrombus formation that can lead to serious vaso-occlusive pathologies like myocardial infarction and ischemic stroke. Despite lack of genomic DNA platelets intriguingly express several transcription factors (*Spinelli et al., 2010*) and developmental morphogens like Wnt (*Kumari and Dash, 2013*; *Steele et al., 2009*) and Sonic Hedgehog (*Kumari et al., 2014*), whose non-canonical non-genomic roles in platelet biology and thrombogenesis remain poorly characterized. Notch signaling has been linked to differentiation of megakaryocytes (*Sugimoto et al., 2006*; *Mercher et al., 2008*), the platelet precursor cells in bone marrow, though there also have been reports to the contrary (*Dorsch et al., 2002*; *Poirault-Chassac et al., 2010*). Here, for the first time, we demonstrate abundant expression of Notch1 and DLL-4, as well as their respective transcripts in human platelets. When platelets were challenged with thrombin, a potent physiological agonist, synthesis and surface translocation of Notch1 and DLL-4 were significantly augmented. Interestingly, DLL-4, in turn, instigated activation of human platelets, as evidenced from binding of PAC-1 and fibrinogen to surface integrins $\alpha_{IIb}\beta_3$, P-selectin externalization, release of adenine nucleotides, shedding of extracellular vesicles (EVs), amplified tyrosine phosphoproteome and rise in intracellular calcium, associated with generation of NICD. In parallel DLL-4 significantly enhanced phosphorylation of PI3K and AKT. Attenuation of γ-secretase significantly abrogated platelet activation responses triggered either by DLL-4 or thrombin. Inhibition of γ-secretase, too, significantly impaired arterial thrombosis in mice and platelet thrombus generation ex vivo. Furthermore, preclusion of DLL-4-Notch1 interaction by pre-incubation with a blocking antibody prohibited thrombin-mediated platelet aggregation and shedding of EVs, which underscores a critical role of Notch signaling in inducing human platelet activation in synergism with physiological agonists in a juxtacrine manner.

## Materials and methods

**Key resources table**

| Reagent type (species) or resource | Designation | Source or reference | Identifiers | Additional information |
|---|---|---|---|---|
| Antibody | Rabbit monoclonal anti-Notch1 | Cell Signaling Technology | 4380 | WB 1:1000 FC 1:100 |
| Antibody | Rabbit polyclonal anti-DLL4 | Novus Biologicals | NB600 892 | WB 1:1000 FC 1:500 Aggregation 2–20 µg/ml |
| Antibody | Rabbit monoclonal anti- cleaved Notch1 | Cell Signaling Technology | 4147 | WB 1:1000 |
| Antibody | Mouse monoclonal anti-phospho(Ser473)-AKT | Cell Signaling Technology | 4051 | WB 1:1000 |
| Antibody | Rabbit polyclonal anti-AKT | Cell Signaling Technology | 9272 | WB 1:1000 |

*Continued on next page*

*Continued*

| Reagent type (species) or resource | Designation | Source or reference | Identifiers | Additional information |
|---|---|---|---|---|
| Antibody | Rabbit polyclonal anti-phospho( Tyr467)-PI3K | Elabscience | E-AB-20966 | WB 1:1000 |
| Antibody | Mouse monoclonal anti-PI3K | Santa Cruz Biotechnology | sc-1637 | WB 1:500 |
| Antibody | Mouse monoclonal anti-p-Tyr | Santa Cruz Biotechnology | sc-7020 | WB 1:5000 |
| Antibody | Rabbit polyclonal anti-actin | Sigma-Aldrich | A2066 | WB 1:5000 |
| Antibody | PE-Mouse anti-human CD62P | BD Biosciences | 550561 | 5 µl |
| Antibody | FITC-Mouse anti-human PAC-1 | BD Biosciences | 340507 | 5 µl |
| Antibody | FITC-Mouse anti-human CD14 | BD Biosciences | 555397 | 10 µl |
| Antibody | APC-Mouse anti-human CD41a | BD Biosciences | 559777 | 10 µl |
| Antibody | DyLight 488-Rat anti-mouse GPIbβ | Emfret Analytics | X488 | 0.1 µg/g body weight |
| Antibody | IgG from rabbit serum | Sigma-Aldrich | I5006 | 10–20 µg/ml |
| Antibody | Alexa Fluor 488-Goat anti-rabbit IgG | Invitrogen | A11008 | 1:100 (Notch1) 1:200 (DLL-4) |
| Antibody | HRP-Goat anti-rabbit IgG | Bangalore Genei | 114038001 A | See Methods- Western analysis |
| Antibody | HRP-Goat anti-mouse IgG | Bangalore Genei | 114068001 A | See Methods- Western analysis |
| Peptide, recombinant protein | DLL-1 | Sino Biological | 11635-H08H | 15 µg/ml |
| Peptide, recombinant protein | DLL-4 | Sino Biological | 10171-H02H | 7.5–15 µg/ml |
| Peptide, recombinant protein | Thrombin receptor-activating peptide (TRAP) | Sigma-Aldrich | S1820 | 2–2.5 µM |
| Peptide, recombinant protein | Bovine serum albumin (BSA) | VWR Life Science | 0332–500 G | |
| Peptide, recombinant protein | Thrombin | Sigma-Aldrich | T6884 | 0.1–1 U/ml |
| Peptide, recombinant protein | Collagen | Chrono-log | 385 | 2–2.5 µg/ml |
| Chemical compound/Inhibitor | N-(N-(3, 5-difluorophenacetyl)-L-alanyl)-*S*-phenyl-glycine *t*-butyl ester (DAPT) | Sigma-Aldrich | D5942 | 10–40 µM |
| Chemical compound/Inhibitor | Dibenzazepine (DBZ) | Selleckchem | YO-01027 | 10–30 µM |
| Chemical compound/Inhibitor | LY-294002 | Sigma-Aldrich | L9908 | 80 µM |
| Chemical compound/Inhibitor | Ro-31–8425 | Calbiochem | 557514 | 20 µM |
| Chemical compound/Inhibitor | Puromycin | Calbiochem | 540222 | 10 mM |
| Chemical compound/Inhibitor | Prostaglandin $E_1$ | Sigma-Aldrich | P5515 | |
| Chemical compound/reagent | Dimethyl sulfoxide (DMSO) | Sigma-Aldrich | D5879 | |
| Chemical compound/reagent | Diethylpyrocarbonate (DEPC) | Amresco | E174 | |
| Chemical compound/reagent | Ethylene glycol tetraacetic acid (EGTA) | Sigma-Aldrich | E-4378 | |
| Chemical compound/reagent | Ethylenediaminetetraacetic acid (EDTA) | Sigma-Aldrich | E9884 | |
| Chemical compound/reagent | $MnCl_2$ | Sigma-Aldrich | M3634 | |
| Chemical compound/reagent | Xylazine | Sigma-Aldrich | X1251 | |
| Chemical compound/reagent | Kaolin | Haemonetics | 6300 | |
| Commercial assay or Kit | Cell Titer-Glo Luminescent Cell Viability Assay Kit | Promega | G7570 | |
| Commercial assay or Kit | Chrono-lume luciferin luciferase reagent | Chrono-log | 395 | |

*Continued on next page*

*Continued*

| Reagent type (species) or resource | Designation | Source or reference | Identifiers | Additional information |
|---|---|---|---|---|
| Commercial assay or Kit | High-capacity reverse transcription kit | Applied Biosystems | 4368814 | |
| Other | Calcein AM | Invitrogen | C3100MP | Cell-permeable dye (2 µg/ml) |
| Other | Fura-2 AM | Calbiochem | 344905 | Cell-permeable dye (2 µM) |
| Other | SYBR Green SuperMix | Bio-Rad | 170–8882 | Dye; see Methods-Quantitative Real-Time PCR |
| Other | TRIzol | Invitrogen | 15596026 | See Methods- RNA extraction |
| Other | Fibrinogen (Alexa Fluor 488-conjugated) | Invitrogen | F13191 | 10 µg/ml |
| Other | Polyvinylidene fluoride (PVDF) membrane | Millipore | IPVH00010 | See Methods- Western analysis |
| Other | Immobilon western chemiluminescent HRP substrate | Millipore | WBKLS0100 | See Methods- Western analysis |
| Other | BD FACS Lysing Solution | BD Biosciences | 349202 | See Methods- Study of platelet-leukocyte interaction |
| Other | Restore Western blot stripping buffer | Thermo Fisher Scientific | 21059 | See Methods- Western analysis |
| Other | Skimmed milk powder | Millipore | 70166 | See Methods- Western analysis |

## Methods

### Study design

No calculations were performed to predetermine sample size. Each experiment was performed independently at least three times or sample size was chosen based on effect size observed during pilot experiments. No inclusion and exclusion criteria were set for experimental units or data points. No outliers were excluded from the analysis. The results reported for all in vitro and ex vivo experiments represent biological replicates (paired observations made on whole blood and/or platelet populations isolated from different healthy volunteers). The results were successfully reproduced with each biological replicate. The results of all in vivo experiments represent independent observations in individual mice. All attempts at reproducing results were successful. All in vitro and ex vivo experiments involved paired observations. No randomization was performed for in vivo experiments. Mice allocated to either control or treatment groups were matched for age, sex and body weight. Other confounding factors were not controlled. Investigators were not blinded to group allocation during data collection and/or analysis.

### Platelet preparation

Platelets were isolated from freshly drawn human blood by differential centrifugation. Briefly, peripheral venous blood collected in acid citrate dextrose (ACD) vial was centrifuged at 100×g for 20 min to obtain platelet-rich plasma (PRP). PRP was then centrifuged at 800×g for 7 min to sediment platelets after adding 1 µM $PGE_1$ and 2 mM EDTA. Pellet was washed with buffer A (20 mM HEPES, 134 mM NaCl, 2.9 mM KCl, 1 mM $MgCl_2$, 0.34 mM $NaH_2PO_4$, 12 mM NaHCO3; pH 6.2) supplemented with 5 mM glucose, 0.35 g/dl BSA and 1 µM $PGE_1$. Finally, platelets were resuspended in buffer B (20 mM HEPES, 134 mM NaCl, 2.9 mM KCl, 1 mM $MgCl_2$, 0.34 mM $NaH_2PO_4$, 12 mM NaHCO3; pH 7.4) supplemented with 5 mM glucose. The final cell count was adjusted to $2–4×10^8$ cells/ml using automated cell counter (Multisizer 4, Beckman Coulter). Leukocyte contamination in platelet preparation was found to be less than 0.015%. All steps were carried out under sterile conditions and precautions were taken to maintain the cells in resting condition. Blood samples were drawn from healthy adult human participants after obtaining written informed consent, strictly as per recommendations and as approved by the Institutional Ethical Committee of the Institute of Medical Sciences, Banaras Hindu

University (Approval No. Dean/2015–16/EC/76). The study methodologies conformed to the standards set by the Declaration of Helsinki.

## Platelet aggregation

Washed human platelets were stirred (12,00 rpm) at 37 °C in a whole blood/optical lumi-aggregometer (Chrono-log model 700–2) for 1 min, followed by addition of agonist (thrombin, TRAP, or collagen) either in presence or absence of reagents. Aggregation was recorded as percent light transmitted through the sample as a function of time, while blank represented 100% light transmission. Platelet aggregation in whole blood, induced by either TRAP or collagen was recorded as change in electrical resistance (impedance) as a function of time.

## Western analysis

Proteins from platelet lysate were separated on 10% SDS-PAGE and electrophoretically transferred onto PVDF membranes by employing either a TE77 PWR semi dry blotter (GE Healthcare) at 0.8 mA/cm$^2$ for 1 hr 45 min or Trans-Blot Turbo Transfer System (Bio-Rad) at 20 V/1.3 A for 30 min (for Notch1 and cleaved Notch1/NICD) or 20 min (for DLL-4, pY99, p-PI3K and pAKT). Membranes were blocked with either 5% skimmed milk or 5% bovine serum albumin in 10 mM Tris HCl, 150 mM NaCl, pH 8.0 containing 0.05% Tween 20 (TBST) for 1 hr at room temperature (RT) to block residual protein binding sites. Membranes were incubated overnight at 4 °C with specific primary antibodies, followed by 3 washings with TBST for 5 min each. Blots were incubated with HRP-conjugated secondary antibodies (goat-anti-rabbit, 1:2500 for anti-Notch1, anti-DLL-4, anti-cleaved Notch1, 1:1000, for p-PI3K, 1:1500 for anti-AKT and 1:40000 for anti-actin; and goat anti-mouse, 1:50000, for anti-pY99, 1:1500, for PI3K, 1:1000 for anti-pAKT) for 1 hr and 30 min at RT, followed by similar washing steps. Antibody binding was detected using enhanced chemiluminescence detection kit (Millipore). Membranes stained for p-PI3K and pAKT were subsequently stripped by incubating in stripping buffer at RT for 30 min, washed, blocked and reprobed employing either anti-PI3K or anti-AKT antibody. Images were acquired on multispectral imaging system (UVP BioSpectrum 800 Imaging System) and quantified using VisionWorks LS software (UVP).

## Analysis of Notch1 and DLL-4 expression on platelet surface

Washed human platelets were stimulated with thrombin (1 U/ml) at 37 °C for 5 min under non-stirring condition. Cells were incubated with either anti-Notch1 antibody (1:100) for 1 hr at RT or anti-DLL-4 antibody (1:500) for 30 min at RT, followed by staining with Alexa Fluor 488-labelled anti-rabbit IgG (1:100, for Notch1; and 1:200, for DLL-4), for 30 min at RT in dark. Cells were washed, resuspended in sheath fluid and were analyzed on a flow cytometer (FACSCalibur, BD Biosciences). Forward and side scatter voltages were set at E00 and 350, respectively, with a threshold of 52 V. An amorphous gate was drawn to encompass platelets separate from noise and multi-platelet particles. All fluorescence data were collected using 4-quadrant logarithmic amplification for 10000 events in platelet gate from each sample and analyzed using CellQuest Pro Software.

## Secretion from platelet α-granules and dense bodies

Secretion from platelet α-granules in response to a stimulus was quantified by surface expression of P-selectin (CD62P). Washed human platelets pre-treated with either DAPT (10 µM) or vehicle for 10 min at RT followed by treatment with either DLL-4 (15 µg/ml) or DLL-1 (15 µg/ml) for 10 min at RT or thrombin (0.1 and 1 U/ml) for 5 min at 37 °C. In other experiments cell were pre-incubated with DLL-4 (7.5 µg/ml) followed by stimulation with thrombin (0.1 U/ml) for 5 min at 37 °C. Cells were stained with PE-labelled anti-CD62P antibody (5 % v/v) for 30 min at RT in dark. Samples were suspended in sheath fluid and subjected to flow cytometry. Secretion of adenine nucleotides from platelet dense granules was measured employing Chrono-lume reagent ( 0.2 µM luciferase/luciferin). Luminescence generated was monitored in a lumi-aggregometer contemporaneous with platelet aggregation (see above). Alternatively, dense granule releasate was quantitated using Cell Titer-Glo Luminescent Cell Viability Assay Kit where cells were sedimented at 800×g for 10 min and supernatant was incubated with equal volume of Cell Titer-Glo reagent for 10 min at RT. Luminescence was recorded in a multimodal microplate reader (BioTeK model Synergy H1).

## Study of platelet integrin activation and fibrinogen binding

Platelet stimulation induces conformational switch in integrins $\alpha_{IIb}\beta_3$ that allows high-affinity binding of fibrinogen leading to cell-cell aggregate formation. Washed human platelets were pre-treated with either DAPT (10 µM), LY-294002 (80 µM) or Ro-31-8425 (20 µM) or vehicle for 10 min at RT followed by exposure to DLL-4 (15 µg/ml) or DLL-1 (15 µg/ml) for 10 min at RT or thrombin (0.1, 0.5 and 1 U/ml) for 5 min at 37 °C. In other experiments cells were pre-incubated with DLL-4 (7.5 µg/ml) followed by stimulation with thrombin (0.1 U/ml) for 5 min at 37 °C. Cells were stained with either FITC-labelled PAC-1 antibody that specifically recognizes active conformation of $\alpha_{IIb}\beta_3$ (5 % v/v) or Alexa Fluor 488-labelled fibrinogen (10 µg/ml) for 30 min at RT in dark. Samples were finally suspended in sheath fluid, and analyzed by flow cytometry.

## Isolation and analysis of platelet-derived extracellular vesicles (PEVs)

PEVs were isolated and characterized as described previously (*Chaurasia et al., 2019*; *Kushwaha et al., 2018*). Platelets were pre-incubated either with DAPT (10 µM) or DBZ (10 µM) for 10 min, followed by treatment with DLL-4 (15 µg/ml) for 10 min at RT. Cells were sedimented at 800×g for 10 min, and then at 1200×g for 2 min at 22 °C° to obtain PEVs cleared of platelets, which were analyzed with Nanoparticle Tracking Analyzer (NTA) where a beam from solid-state laser source (635 nm) was allowed to pass through the sample. Light scattered by rapidly moving particles in suspension in Brownian motion at RT was observed under 20 X microscope. This revealed hydrodynamic diameters of particles, calculated using Stokes Einstein equation, within range of 10 nm to 1 µm and concentration between $10^7$ and $10^9$ /ml. The average distance moved by each EV in x and y directions were captured with CCD camera (30 frames/s) attached to the microscope. Both capture and analysis were performed using NanoSight LM10 (Malvern) and NTA 2.3 analytical software, which provide an estimate of particle size and counts in sample.

## Measurement of intracellular free calcium

Intracellular calcium was measured as described (*Chaurasia et al., 2019*). Briefly, platelet-rich plasma (PRP) was isolated from fresh human blood and incubated with Fura-2 AM (2 µM) at 37 °C for 45 min in dark. Fura-2 labelled platelets were isolated, washed and finally resuspended in buffer B. Fluorescence for each sample was recorded in 400 µl aliquots of platelet suspensions at 37 °C under non-stirring condition by Hitachi fluorescence spectrophotometer (model F-2500). Excitation wavelengths were 340 and 380 nm and emission wavelength was set at 510 nm. Changes in intracellular free calcium concentration, $[Ca^{2+}]_i$, was monitored from fluorescence ratio (340/380) using Intracellular Cation Measurement Program in FL Solutions software. $F_{max}$ was determined by lysing the cells with 40 µM digitonin in presence of saturating $CaCl_2$. $F_{min}$ was determined by the addition of 2 mM EGTA. Intracellular free calcium was calibrated according to the derivation of *Grynkiewicz et al., 1985*.

## Study of platelet-leukocyte interaction

Fresh human blood (20 µl) was added to a cocktail containing 10 µl each from APC-anti-CD41a (platelet-specific) and FITC-anti-CD14 (leukocyte-specific) antibodies and mixed gently. Samples were treated with either DAPT (40 µM) or vehicle for 10 min, followed by incubation with either TRAP (2 µM) or DLL-4 (15 µg/ml) for 15 min at RT. RBCs were lysed with 800 µl FACS lysis solution (1 X, BD Biosciences) for 10 min at RT. Platelet-leukocyte interaction was analyzed on a flow cytometer. Side scatter voltage was set at 350 with a threshold of 52 V and amorphous gates were drawn to encompass neutrophils and monocytes separate from noise. A dot plot of side scatter (SSC) versus log FITC-CD14 fluorescence was created in the CellQuest Pro software. Amorphous gates were drawn for monocyte (high fluorescence and low SSC) and neutrophil (low fluorescence and high SSC) populations. All fluorescence data were collected using 4-quadrant logarithmic amplification for 1000 events in either neutrophil or monocyte gate from each sample and analyzed using CellQuest Pro Software.

## Thromboelastography (TEG)

Coagulation parameters in whole blood were studied by employing Thromboelastograph 5000 Hemostasis Analyzer System (Haemonetics) and TEG analytical software. Whole blood (1 ml) was incubated either with DAPT (20 µM) or vehicle for 10 min at RT, followed by transfer to citrated kaolin tubes

with proper mixing. CaCl₂ (20 μl) was added to 340 μl sample to initiate coagulation cascade. Mixture was placed in disposable TEG cups and data were collected as per to manufacturer instructions until maximum amplitude was reached or 60 min had elapsed.

## Intravital imaging of thrombus formation in murine mesenteric arterioles

Ferric chloride-induced mesenteric arteriolar thrombosis in mice was imaged by intravital microscopy as previously described (*Kulkarni et al., 2019*; *Chaurasia et al., 2019*) with minor modifications. The animal study was ethically approved by the Central Animal Ethical Committee of Institute of Medical Sciences, Banaras Hindu University (Approval No. Dean/2017/CAEC/83). All efforts were made to minimize the number of animals used, and their suffering. Mice (species: *Mus musculus*; strain: Swiss albino; sex: male and female; age: 4-5 weeks old; weight: 8-10 g each) were anaesthetized with intra-peritoneal injection of ketamine/xylazine cocktail (100 mg/kg ketamine and 10 mg/kg xylazine). Anti-GPIbβ antibody (DyLight 488-labelled, 0.1 μg/g body weight) diluted in 50 μl sterile PBS was injected into retro-orbital plexus of mice in order to fluorescently label circulating platelets. Mesentery was exposed through a mid-line incision in abdomen and kept moist by superfusion with warm (37 °C) sterile PBS. An epifluorescence inverted video microscope (Nikon model Eclipse Ti-E) equipped with monochrome CCD cooled camera was employed to image isolated mesenteric arterioles of diameter 100-150 μm. The arteriole was injured by topically placing a Whatman filter paper saturated with ferric chloride (10%) solution for 3 min and thrombosis in the injured vessel was monitored in real time for 40 min or until occlusion. Movies were subsequently analyzed with Nikon image analysis software (NIS Elements) to determine (a) the time required for formation of first thrombus (>20 μm in diameter), (b) time required for occlusion of the vessel i.e. time required after injury till stoppage of blood flow for 30 s, and (c) thrombus growth rate i.e. growth of a thrombus (>30 μm diameter) followed over a period of 3 min. Fold increase was calculated by dividing diameter of thrombus at given time (n) by the diameter of the same thrombus at time (0). Time 0 was defined as the time point at which thrombus diameter first reached the size 30 μm approximately.

## Study of platelet thrombus formation on immobilized collagen matrix under arterial shear

Platelet adhesion and thrombus growth on immobilized collagen matrix was quantified by using BioFlux (Fluxion Biosciences) microfluidics system as described previously (*Sonkar et al., 2019*). Wells of high-shear plates were coated with 50 μl collagen (from 100 μg/ml stock) at 10 dynes/cm² for 30 s and were left for 1 hr at RT. Wells were blocked with 1% bovine serum albumin at 10 dynes/cm² for 15 min at RT. Platelets stained with Calcein AM (2 μg/ml) were perfused over collagen at physiological arterial shear rate (1500 sec⁻¹) for 5 min. Adhesion of platelets and thrombus formation in a fixed field over time was recorded. Representative images from 5 to 10 different fields were captured and total area occupied by thrombi at 5 min in 5 representative fields was analyzed using ImageJ software (National Institutes of Health).

## Quantitative real-time PCR

### RNA extraction

Platelets were isolated from human blood as described above. Precaution was taken to prevent leukocyte contamination. Cells were counted with Beckman Coulter Counter Multisizer 4. Total RNA extraction, reverse transcription and qRT-PCR were carried out as described (*Kumari et al., 2015*). Total RNA was extracted from platelets using TRIzol reagent according to the protocol of the manufacturer and suspended in DEPC-treated water.

### Reverse transcription

Platelet RNA (1 μg) was transcribed to cDNA using a high-capacity cDNA reverse transcription kit (Applied Biosystems) according to the instructions of the manufacturer. Samples were amplified in a PTC-150 thermal cycler (MJ Research) by using the following program: 25 °C for 10 min, 37 °C for 2 hr, and 85 °C for 5 min.

**Table 1.** Details of primers employed in amplification reactions.

| Genes | Forward Primers (5' to 3') | Reverse Primers (5' to 3') | Amplicon Size (bp) | Annealing Temp (°C) |
|---|---|---|---|---|
| GAPDH | GAAGGTGAAGGTCGGAGTC | GAAGATGGTGATGGGATTTC | 226 | 57 |
| ACTB | AAATCTGGCACCACACCTTC | AGCACAGCCTGGATAGCAAC | 160 | 59 |
| NOTCH1 | TCAGCGGGATCCACTGTGAG | ACACAGGCAGGTGAACGAGTTG | 104 | 62 |
| NOTCH2 | TGCCAAGCTCAGTGGTGTTGTA | TGCTAGGCTTTGTGGGATTCAG | 132 | 60 |
| NOTCH3 | GGTTCCCAGTGAGCACCCTTAC | GTGGATTCGGACCAGTCTGAGAG | 100 | 60 |
| NOTCH4 | CGGCCTCGGACTCAGTCA | CAACTCCATCCTCATCAACTTCTG | 112 | 60 |
| DLL1 | TGTGTGACGAACACTACTACGGAG | GTGAAGTGGCCGAAGGCA | 76 | 65 |
| DLL3 | GAGACACCCAGGTCCTTTGA | CAGTGGCAGATGTAGGCAGA | 61 | 65 |
| DLL4 | CCAGGAAAGTTTCCCCACAGT | CCGACACTCTGGCTTTTCACT | 82 | 65 |
| JAG1 | GCTGGCAAGGCCTGTACTG | ACTGCCAGGGCTCATTACAGA | 78 | 65 |
| JAG2 | CACCGAGGTCAAGGTGGAGA | ACGCTGAAGGCACCACACA | 84 | 65 |

## Quantitative real-Time PCR

Primers were designed using the latest version of Primer3 input software. The primers (forward and reverse) for target genes were obtained from Eurofins Genomics and presented in *Table 1*. Glyceraldehyde 3-phosphate dehydrogenase (*GAPDH*) and *ACTB* were used as the reference genes. We performed real-time PCR employing SYBR Green SuperMix in a CFX-96 real-time PCR system (Bio-Rad). Thermal cycling conditions were as follows: 95 °C for 3 min, followed by 40 cycles consisting of 10 s of denaturation at 95 °C, 10 s of annealing (at temperatures mentioned in the *Table 1*), and extension at 72 °C. A melt peak analysis of amplicons was carried out to rule out nonspecific amplifications.

## Statistical methods

Standard statistical methods were employed in the study. Two tailed Student's *t* test (paired or unpaired) (for two groups) or RM one-way analysis of variance (ANOVA) (for more than two groups) with either Dunnett's or Sidak's multiple comparisons test was used for evaluation. Tests were considered significant at $p < 0.05$. All the analysis was carried out employing GraphPad Prism version 8.4. Linear regression analysis was performed for in vivo studies, and the slopes from best-fit were used to arrive at rates in time-lapse experiments. Kalpan-Meier analysis and Log-Rank test were performed to determine significance of difference in time to occlusion of vessel between different groups. Data are presented as mean ± SEM of at least three individual experiments.

## Results

### Notch1 and DLL-4 are abundantly expressed in human platelets

Although enucleate, platelets inherit a limited transcriptome from precursor megakaryocytes (*Freedman, 2011*; *McRedmond et al., 2004*). Notch1 is a transmembrane protein present on cell surfaces and is part of a highly conserved Notch signaling pathway (*van Tetering et al., 2011*). We searched for the expression of transcripts of Notch isoforms and its ligands in platelets by RT-qPCR. The Cq values for housekeeping genes (*GAPDH* and *ACTB*) were determined as 21 and 23, respectively, whereas that for *NOTCH1* was 27 (*Figure 1—figure supplement 1*, A and B), which was reflective of abundant expression of *NOTCH1* mRNA in human platelets. Contrasting this, *NOTCH* isoforms 2, 3 and 4 had Cq values greater than 33 (*Figure 1—figure supplement 1*, A and B). Keeping with above, there was notable existence of Notch1 peptide in human platelets, whose level significantly increased upon stimulation with thrombin (1 U/ml), a potent physiological agonist (*Figure 1*, A and B). Pre-treatment of platelets with puromycin (10 mM) singnificantly deterred synthesis of this pepetide (*Figure 1*, A and B). We also observed considerable expression of Notch1 on platelet surface membrane, whose level enhanced significantly (by 65.71%) upon thrombin-stimulation (*Figure 1*, C and D).

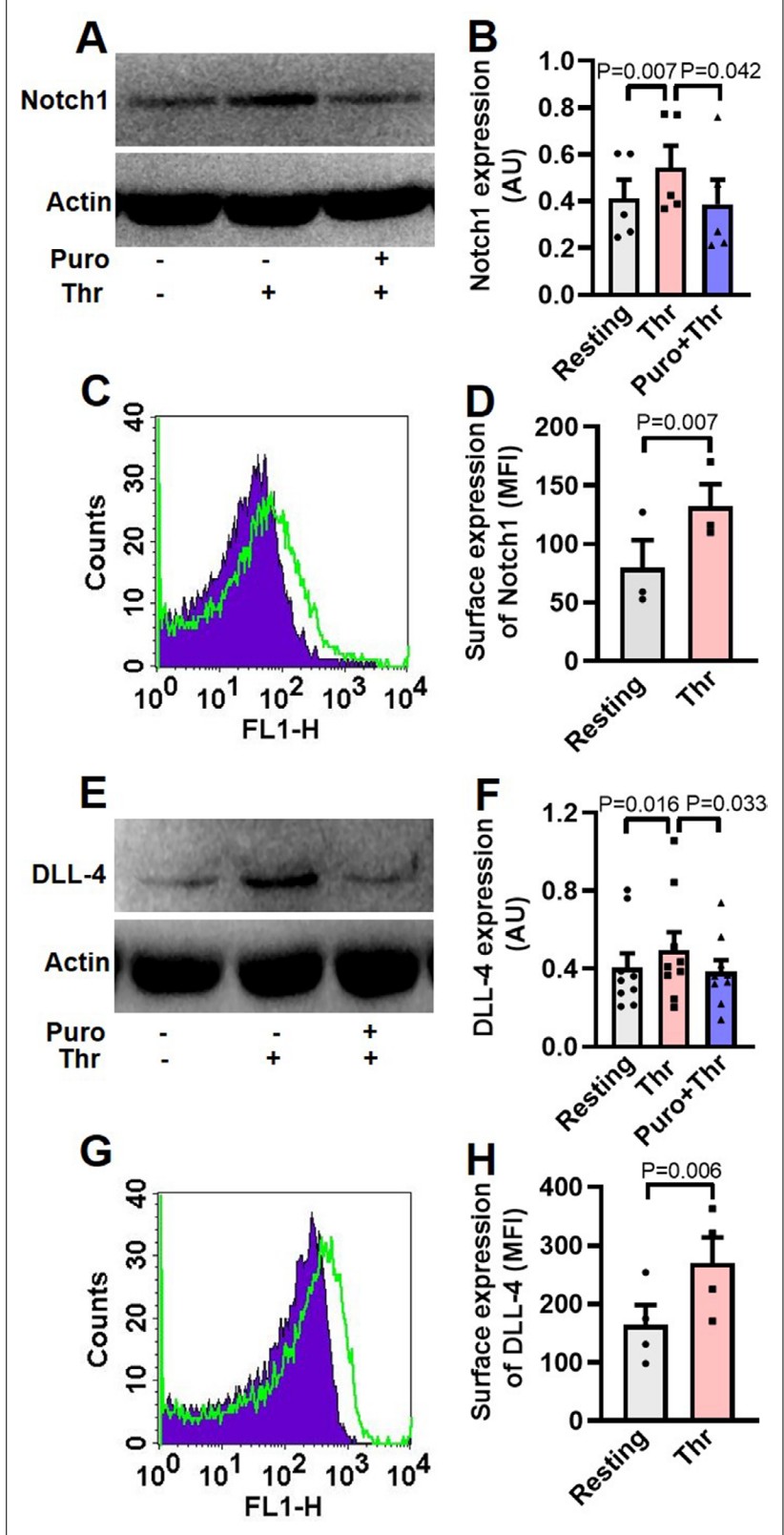

**Figure 1.** Human platelets express Notch1 and DLL-4. (**A**) Immunoblot demonstrating expression of Notch1 in platelets pre-treated with or without puromycin (Puro, 10 mM), followed by stimulation with thrombin (Thr, 1 U/ml, for 5 min at 37 °C). (**B**) Corresponding densitometric analysis of Notch1 normalised with β-actin (n=5). (**C**) Flow cytometric analysis of platelets treated with (unshaded) or without (shaded) thrombin (1 U/ml) for 5 min

*Figure 1 continued on next page*

*Figure 1 continued*

at 37 °C, followed by staining with anti-Notch1 antibody and Alexa Fluor 488-labelled secondary antibody. (**D**) Corresponding mean fluorescence intensity (MFI) of Notch1 expression on platelets as indicated (n=3). (**E**) Immunoblot showing synthesis of DLL-4 in thrombin-stimulated platelets. (**F**) Corresponding densitometric analysis of DLL-4 normalised with β-actin (n=9). (**G**) Histogram showing expression of DLL-4 on surface of human platelets pre-treated with (unshaded) or without (shaded) thrombin (1 U/ml) for 5 min at 37 °C, followed by incubation with anti-DLL-4 antibody and Alexa Fluor 488-labelled secondary antibody. (**H**) Corresponding mean fluorescence intensity of DLL-4 expression on platelets as indicated (n=4). Data are presented as mean ± SEM of at least three different experiments. Analyzed by either Student's paired *t*-test (**D** and **H**) or RM one-way ANOVA with Dunnett's multiple comparisons test (**B** and **F**).

The online version of this article includes the following source data and figure supplement(s) for figure 1:

**Source data 1.** Excel sheet shows numerical data of *Figure 1*.

**Source data 2.** Unedited and labelled blots of *Figure 1*.

**Source data 3.** Unedited and unlabelled blots of *Figure 1*.

**Figure supplement 1.** Human platelets abundantly express *NOTCH1* mRNA.

**Figure supplement 1—source data 1.** Excel sheet shows critical quantity (Cq) values of *Figure 1—figure supplement 1*.

**Figure supplement 2.** Human platelets abundantly express *DLL4* mRNA.

**Figure supplement 2—source data 1.** Excel sheet shows critical quantity (Cq) values of *Figure 1—figure supplement 2*.

The Cq of *DLL4*, *JAG1*, and *JAG2* were found to be 26, 31 and 30, respectively while those for *DLL* isoforms-1 and -3 were higher than or equal to 33 (*Figure 1—figure supplement 2A and B*), reflective of *DLL4* being the most abundantly expressed Notch ligand transcript in human platelets. Melt peak analyses were supportive of lack of formation of by-products (*Figure 1—figure supplement 1C* and *Figure 1—figure supplement 2C*). Consistent with above, platelets were found to express DLL-4 peptide whose level increased significantly when cells were challenged with thrombin (1 U/ml) (*Figure 1E and F*). Rise in DLL-4 could be averted upon pre-incubation of platelets with puromycin (10 mM) (*Figure 1E and F*). Thrombin, too, significantly augmented surface translocation of DLL-4 by 64.31% (*Figure 1G and H*), thus raising possibility of DLL-4-Ntoch1 interaction on adjacent platelet membranes. As enucleate platelets are known to have limited capacity for protein synthesis, the present observations add Notch1 and DLL-4 to the growing list of platelets translatome.

## DLL-4 amplifies expression of Notch intracellular domain (NICD) in human platelets

Interaction of Notch1 with cognate ligands leads to sequential cleavage of the transmembrane receptor and generation of NICD (*Iso et al., 2003*). As Notch1 is expressed in human platelets, we asked whether exposure to DLL-4 would evoke release of NICD in these cells. Remarkably, exposure of platelets with DLL-4 (15 µg/ml) for 10 min led to significant rise (by 5.1-fold) in level of NICD (*Figure 2A and B*). As NICD generation is mediated through activity of γ-secretase, we next investigated the contribution of this protease in DLL-4-induced NICD release in platelets. Pre-treatment of platelets with either DAPT (10 µM) or DBZ (10 µM), specific inhibitors of γ-secretase, for 10 min led to significant drop in DLL-4-induced NICD release (by 25.33% and 23.77%, respectively) (*Figure 2A and B*), strongly suggestive of functional DLL-4-Notch1-NICD signaling axis in human platelets.

Interestingly, level of NICD was reduced by 2.4, 43.4, 70.3, and 84.9%, respectively, when platelets were stored for 1, 3, 5, and 8 hr at 37 °C in presence of 1 mM calcium (*Figure 2C*). However, NICD level was not considerably affected upon storage of cells at 22 °C. As calpain, the $Ca^{2+}$-dependent thiol protease, is known to be activated in platelets stored at 37 °C, and not at 22 °C (*Wadhawan et al., 2004*), we pre-incubated cells at 37 °C with either calpeptin (80 µM) or ALLN (50 µM), specific inhibitors of calpain, or divalent ion chelator EGTA (1 mM). Significant recovery of NICD intensity under above conditions (*Figure 2D*) was consistent with NICD being a calpain substrate. In keeping with this observation, incubation of platelets with calcium ionophore A23187 (1 µM) for 10 min at 37 °C in presence of 1 mM calcium brought about significant reduction (by 29.77%) in the level of NICD, which was restored upon pre-treatment with either of the calpain inhibitors (*Figure 2E*).

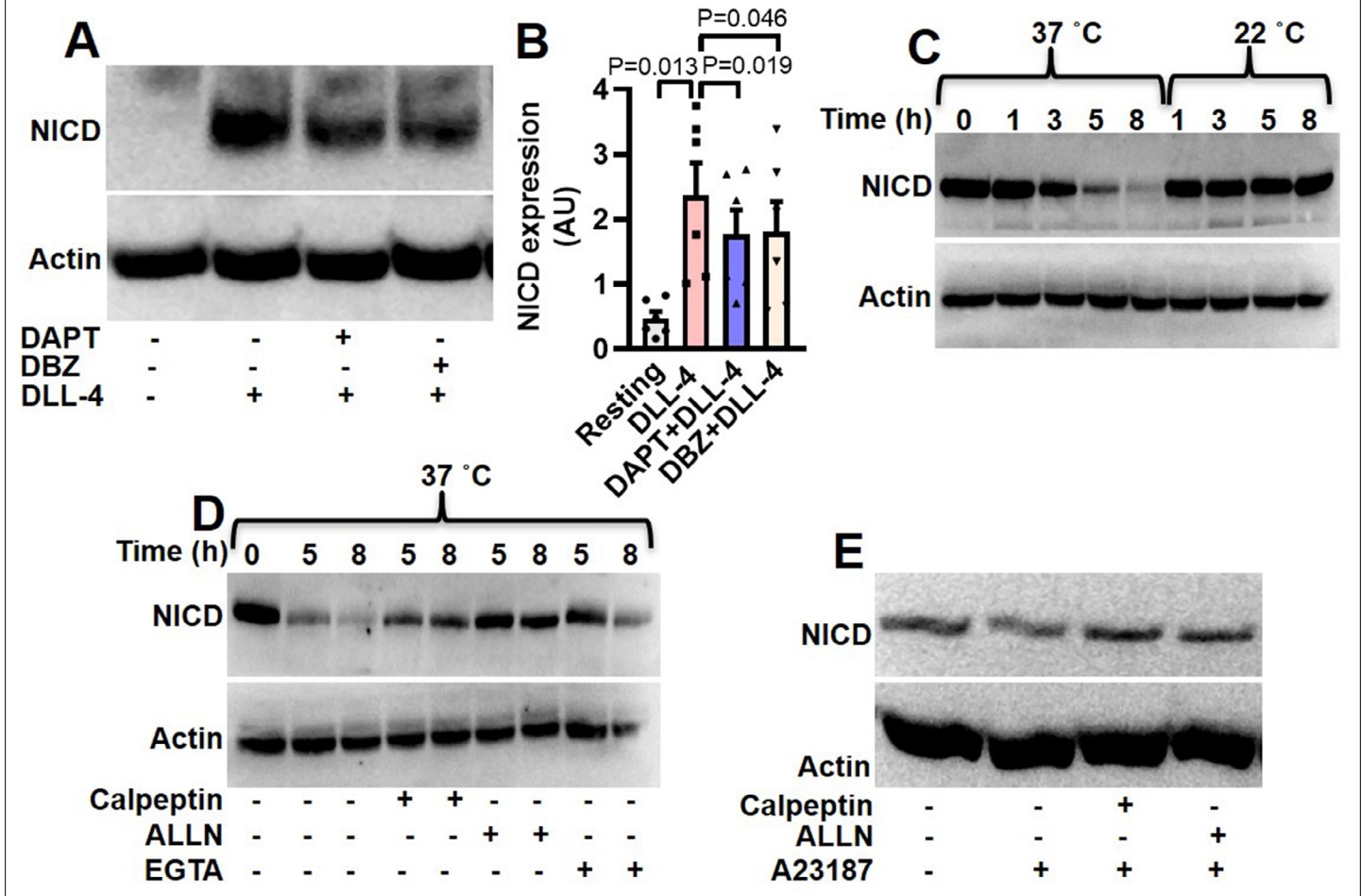

**Figure 2.** Expression of NICD in human platelets. (**A**) Immunoblot showing expression of NICD in DLL-4 (15 µg/ml for 10 min)-treated platelets in absence or presence of either DAPT (10 µM) or DBZ (10 µM) or vehicle. (**B**) Corresponding densitometric analysis of NICD normalised with β-actin (n=6). (**C**, **D** and **E**) Immunoblot of NICD expression in either stored or A23187 (1 µM)-treated platelets under conditions as indicated. Data are represented as mean ± SEM of at least three individual experiments and analyzed by RM one-way ANOVA with Dunnett's multiple comparisons test.

The online version of this article includes the following source data for figure 2:

**Source data 1.** Excel sheet shows numerical data of *Figure 2*.

**Source data 2.** Unedited and labelled blots of *Figure 2*.

**Source data 3.** Unedited and unlabelled blots of *Figure 2*.

## DLL-4 but not DLL-1 induces integrin $\alpha_{IIb}\beta_3$ activation, exocytosis of granule contents, rise in intracellular calcium, extracellular vesicle shedding, platelet-leukocyte aggregate formation and increase in tyrosine phosphoproteome in human platelets

Hallmark of activated platelets is the conformational switch of its surface integrins $\alpha_{IIb}\beta_3$ that allows high-affinity binding of fibrinogen, associated with release of granule contents, rise in intracellular free calcium and shedding of extracellular vesicles. To study the effect of Notch ligands we pre-incubated platelets with DLL-4 (15 µg/ml for 10 min at RT) that prompted enhanced binding of PAC-1-FITC (that recognizes the open conformation of $\alpha_{IIb}\beta_3$) (*Figure 3*, A and B) and fibrinogen-Alexa Fluor 488 (*Figure 3—figure supplement 1*) by 3.07- and 3.13- folds, respectively. DLL-1 was notably ineffective in eliciting such response. Furthermore, platelets exposed to DLL-4 were found to have significant surface expression of P-selectin as a measure of α-granule secretion while no change was observed with DLL-1 (*Figure 3*, C and D). In keeping with above, DLL-4 also induced release of ATP from platelet dense granules (*Figure 3E*). Although DLL-4, on its own, did not incite platelet aggregation at the

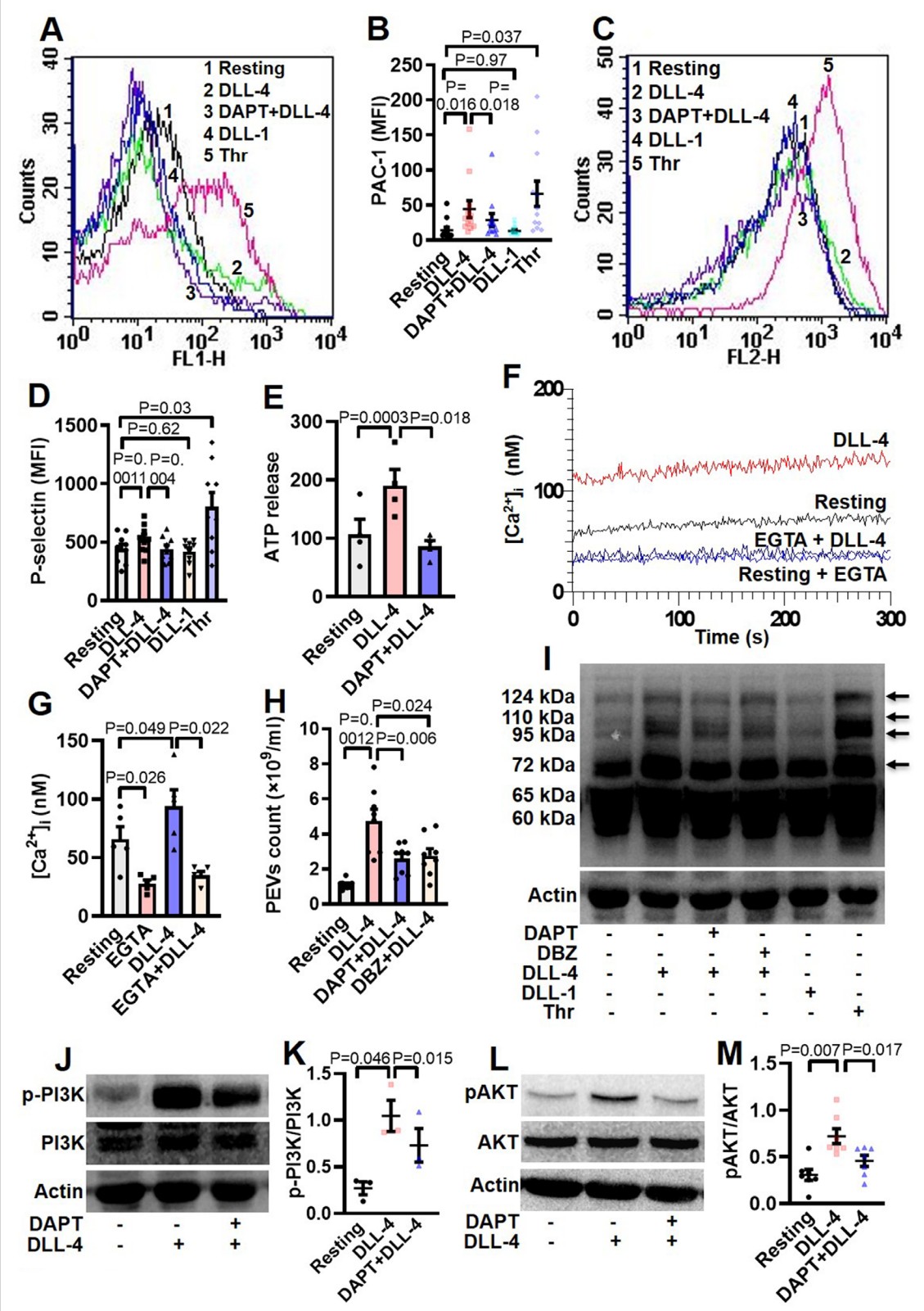

**Figure 3.** DLL-4 induces integrin activation, P-selectin externalization, ATP release, extracellular vesicle shedding, rise in intracellular Ca²⁺ and increase in tyrosine phosphoproteome in human platelets. (**A** and **C**) Histograms showing binding of PAC-1 (**A**) and anti-P-selectin antibody (**C**) to platelets pre-incubated with either DAPT (10 μM) or vehicle for 10 min at RT followed by treatment with either DLL-4 (15 μg/ml) or DLL-1 (15 μg/ml) for 10 min, or with thrombin (Thr, 1 U/ml) for 5 min at 37 °C as indicated. (**B** and **D**) Ccorresponding mean fluorescence intensities of PAC-1 (n=12) and anti-P-selectin

*Figure 3 continued on next page*

*Figure 3 continued*

antibody (n=9) binding to platelets, respectively. (**E**) Bar diagram representing ATP secretion from platelet dense granules pre-incubated with either DAPT (10 µM) or vehicle for 10 min at RT followed by treatment with DLL-4 for 10 min (n=4). (**F**) Fura-2-loaded platelets were pre-treated for 5 min either with calcium (1 mM) or EGTA (1 mM) followed by incubation with DLL-4 (15 µg/ml) for 15 min and intracellular $Ca^{2+}$ was measured. (**G**) Corresponding bar diagram representing mean concentration of intracellular $Ca^{2+}$ over 300 sec of measurement (n=5). (**H**) Platelets were pre-treated with either DAPT (10 µM) or DBZ (10 µM) or vehicle for 10 min at RT followed by treatment with DLL-4 (15 µg/ml) for 10 min at RT. PEVs were isolated and analyzed with Nanoparticle Tracking Analyzer (n=8). (**I**) Immunoblot showing profile of tyrosine phosphorylated proteins in platelets pre-treated with either DAPT (10 µM) or DBZ (10 µM) or vehicle for 10 min at RT followed by treatment with either DLL-4 (15 µg/ml) for 10 min at RT or DLL-1 (15 µg/ml) for 10 min at RT or with thrombin (1 U/ml) for 5 min at 37 °C as indicated (n=4). Arrows indicate position of peptides whose intensity increased in presence of DLL-4. (**J** and **L**) Immunoblots showing expression of p-PI3K and pAKT in DLL-4 (15 µg/ml for 10 min)-treated platelets in absence or presence of either DAPT (10 µM) or vehicle. (**K** and **M**) Corresponding densitometric analyses normalised with PI3K (n=3) and AKT (n=7), respectively. Data are presented as mean ± SEM of at least three different experiments. Results were analyzed by RM one-way ANOVA with either Dunnett's multiple comparisons test (**E**, **H**, **K** and **M**) or Sidak's multiple comparisons test (**B**, **D** and **G**).

The online version of this article includes the following source data and figure supplement(s) for figure 3:

Source data 1. Excel sheet shows numerical data of *Figure 3*.

Source data 2. Unedited and labelled blots of *Figure 3*.

Source data 3. Unedited and unlabelled blots of *Figure 3*.

Figure supplement 1. DLL-4 induces fibrinogen binding to human platelets.

Figure supplement 1—source data 1. Excel sheet shows numerical data of *Figure 3—figure supplement 1*.

Figure supplement 2. DLL-4 induces platelet-leukocyte aggregate formation.

Figure supplement 2—source data 1. Excel sheet shows numerical data of *Figure 3—figure supplement 2*.

Figure supplement 3. Inhibitors of PI3K and PKC prevent PAC-1 binding to DLL-4-induced human platelets.

Figure supplement 3—source data 1. Excel sheet shows numerical data of *Figure 3—figure supplement 3*.

doses employed, it could significantly potentiate thrombin-mediated platelet aggregation (*Figure 4*, A and B).

As Notch signaling is mediated through activity of γ-secretase leading to cleavage of Notch receptor, we next investigated the role of this protease in DLL-4-induced platelet activation. Platelets were pre-treated with DAPT (10 µM), a specific γ-secretase inhibitor, for 10 min at RT followed by exposure to DLL-4. Interestingly, we observed significant drop in DLL-4-induced activation of integrin $\alpha_{IIb}\beta_3$ (*Figure 3*, A and B; *Figure 3—figure supplement 1*), P-selectin exposure (*Figure 3*, C and D) and release of ATP from platelet dense granules (*Figure 3E*) when platelets were pre-incubated with DAPT.

P-selectin expressed on stimulated platelets serves as a ligand for P-selectin glycoprotein ligand-1 (PSGL-1) receptor on leukocytes leading to platelet-leukocyte interaction. As DLL-4 incited P-selectin exposure on platelet surface, we asked next whether it would, too, prompt interaction between the two cell types. Remarkably, addition of DLL-4 (15 µg/ml, 10 min) to fresh human blood led to significant boost in platelet-neutrophil and platelet-monocyte aggregates, which was reduced upon pre-treatment with DAPT (40 µM, 10 min) (*Figure 3—figure supplement 2*). Above observations underline a critical role of Notch signaling in platelet-leukocyte interaction and thrombogenesis.

Rise in intracellular $Ca^{2+}$, $[Ca^{2+}]_i$, is a hallmark of stimulated platelets (*Mallick et al., 2015*). We next determined the possible effect of DLL-4 on calcium flux in human platelets. Interestingly, exposure to DLL-4 (15 µg/ml) for 10 min evoked significant rise (by 1.34-fold) in $[Ca^{2+}]_i$ in Fura-2 AM-stained platelets in presence of 1 mM extracellular $Ca^{2+}$ (*Figure 3*, F and G). To validate whether calcium entry from external medium contributed to rise in $[Ca^{2+}]_i$, we pre-treated cells with EGTA (1 mM) followed by incubation with DLL-4. Chelation of extracellular calcium led to significant drop in rise in $[Ca^{2+}]_i$ (by 64.87%), suggestive of DLL-4-mediated $Ca^{2+}$ influx in these platelets (*Figure 3*, F and G).

Platelet-derived extracellular vesicles (PEVs) are cellular fragments ranging in size between 0.1 and 1 µm that are shed by activated platelets (*Kulkarni et al., 2019*; *Heijnen et al., 1999*). PEVs are pro-coagulant in nature that significantly contribute to haemostatic responses (*Sinauridze et al., 2007*; *Mallick et al., 2015*). Exposure of platelets to DLL-4 (15 µg/ml) for 10 min led to extensive shedding of PEVs, which were 4.29-fold higher in count than those released from vehicle-treated counterparts (*Figure 3H*). Interestingly, pre-treatment of platelets with either DAPT (10 µM) or DBZ (10 µM), specific

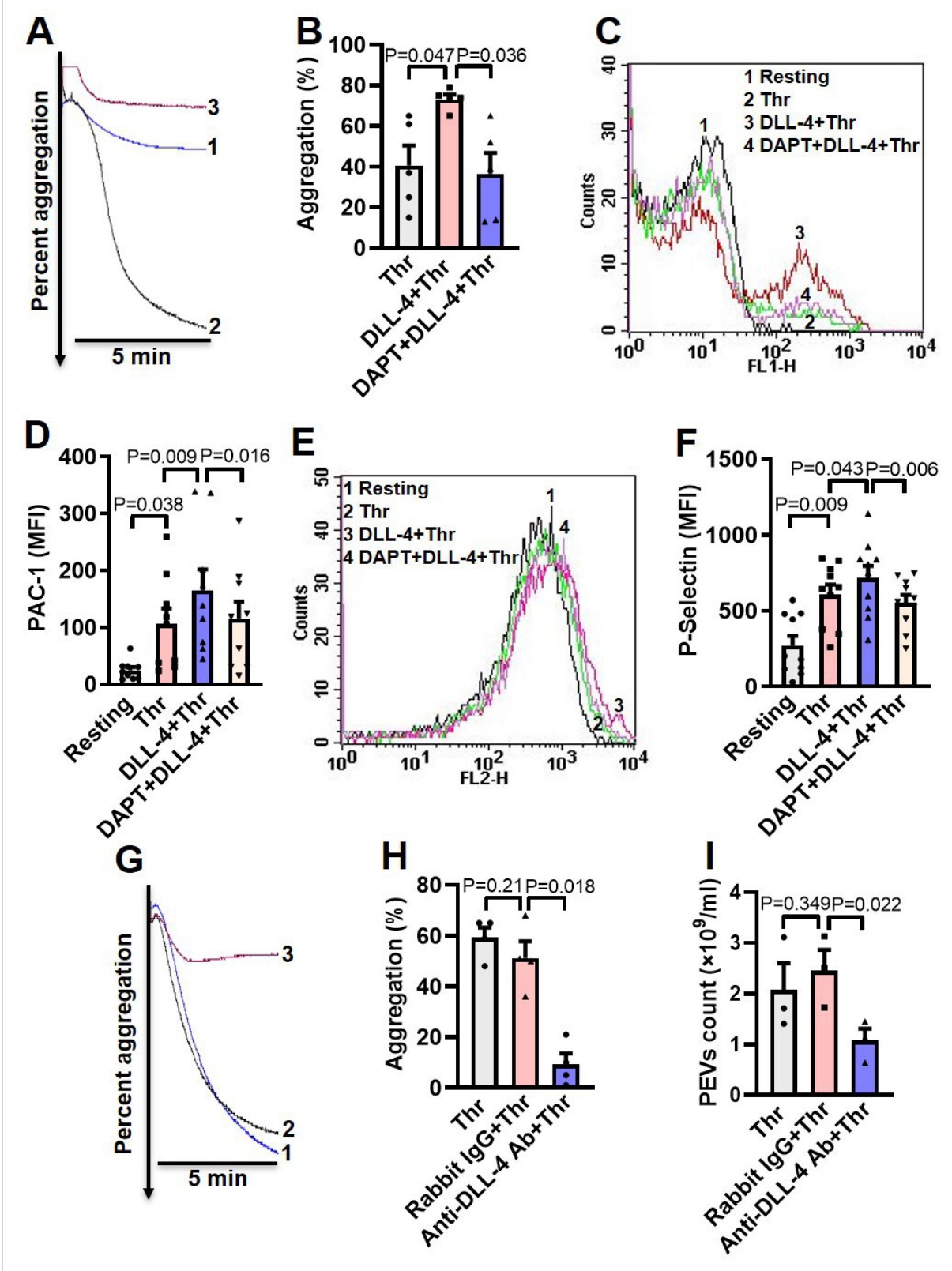

**Figure 4.** DLL-4 operates in a juxtacrine manner to potentiate thrombin-mediated platelet activation. (**A**) Aggregation of washed human platelets induced by thrombin (Thr, 0.1 U/ml) either in presence of vehicle (tracing 1) or DLL-4 (15 μg/ml, tracing 2). Tracing 3 represents cells pre-incubated with DAPT (20 μM) for 10 min at RT followed by addition of DLL-4 and thrombin. (**B**) Corresponding bar chart representing mean platelet aggregation (n=5). (**C** and **E**) Histograms representing PAC-1 binding (**C**) and surface expression of P-selectin (**E**) in platelets pre-treated with DLL-4 (7.5 μg/ml) for 10 min

*Figure 4 continued on next page*

*Figure 4 continued*

followed by stimulation with thrombin (0.1 U/ml) as indicated. Tracings 4 of **C** and **E** represent cells pre-incubated with DAPT (10 µM) for 10 min at RT followed by addition of DLL-4 and thrombin. (**D** and **F**) corresponding mean fluorescence intensity of PAC-1 binding (n=9) and surface expression of P-selectin (n=10), respectively. (**G**) Aggregation of washed human platelets induced by thrombin (0.1 U/ml) following pre-treatment with either rabbit-IgG (20 µg/ml) for 5 min (tracing 2), or anti-DLL-4 antibody (20 µg/ml) for 5 min (tracing 3) or vehicle (tracing 1). (**H**) Corresponding bar chart representing mean platelet aggregation (n=4). (**I**) Platelets were pre-treated with either anti-DLL-4 antibody (20 µg/ml) or rabbit IgG (20 µg/ml) or vehicle for 5 min at RT followed by aggregation induced by thrombin (0.1 U/ml) for 5 min at 37°C. EVs were isolated from aggregated platelets and analyzed with Nanoparticle Tracking Analyzer (n=3). Data are representative of at least three different experiments and presented as mean ± SEM. Analyzed by RM one-way ANOVA with either Dunnett's multiple comparisons test (**B**, **H**, and **I**) or Sidak's multiple comparisons test (**D** and **F**).

The online version of this article includes the following source data and figure supplement(s) for figure 4:

**Source data 1.** Excel sheet shows numerical data of *Figure 4*.

**Figure supplement 1.** Anit-DLL-4 antibody inhibits thrombin-mediated platelet aggregation in a dose-dependent manner.

**Figure supplement 1—source data 1.** Excel sheet shows numerical data of *Figure 4—figure supplement 1*.

γ-secretase inhibitors, for 10 min led to significant drop in DLL-4-induced PEVs release (by 45.55% and 41.98%, respectively) (*Figure 3H*), thus underscoring critical role of γ-secretase activity.

Platelet activation is associated with phosphorylation of multiple cytosolic proteins on tyrosine residues (*Golden et al., 1990*). Platelets treated with DLL-4 but not DLL-1 evoked increased tyrosine phosphorylation of peptides having Mr 72, 95, 110, and 124 kDa, which was remarkably reduced in presence of DAPT and DBZ. Above observations are indicative of DLL-4-γ-secretase axis-induced flux in tyrosine phosphoproteome in human platelets (*Figure 3I*). Thrombin-stimulated platelets were employed in the study as positive control.

Roles of phosphatidylinositol (PI) 3-kinase and protein kinase C (PKC) in platelet activation have been widely reported (*Hirsch et al., 2001*; *Polanowska-Grabowska and Gear, 1999*; *Atkinson et al., 2001*; *Watson and Hambleton, 1989*). In order to implicate these kinases in DLL-4-mediated integrin $\alpha_{IIb}\beta_3$ activation, platelets were pre-treated with either LY-294002 (80 µM) or Ro-31–8425 (20 µM), inhibitors of PI3K and PKC, respectively, or vehicle for 10 min at RT, followed by incubation with DLL-4 (15 µg/ml) for 10 min. Strikingly, both the inhibitors triggered significant drop in DLL-4-induced PAC1 binding to platelets (*Figure 3—figure supplement 3*), which underscored the roles of PI3K and PKC in DLL-4-induced conformational changes in integrins $\alpha_{IIb}\beta_3$. Consistent with activation of PI3K, phosphorylation of its p85 regulatory subunit was significantly augmented (by 3.85-fold) in presence of DLL-4 (15 µg/ml, 10 min), which, too, provoked significant upregulation (by 2.35-fold) in phosphorylation of AKT, the enzyme downstream of PI3K (*Figure 3*, J-M). The phosphorylations of PI3K as well as AKT were significantly attenuated by 30.05% and 36.67%, respectively, upon pre-treatment with DAPT (10 µM for 10 min) (*Figure 3*, J-M). These findings are strongly suggestive of non-canonical signaling evoked by DLL-4 in human platelets in γ-secretase-dependent manner leading to platelet activation.

## DLL-4 operates in a juxtacrine manner to potentiate thrombin-mediated platelet activation

As thrombin triggers synthesis and expression of DLL-4 on platelet surface, which, in turn, induces platelet activation signaling, we asked next whether DLL-4 synergizes with thrombin in transforming platelets to 'pro-active / pro-thrombotic' phenotype. Interestingly, there was significant upregulation in platelet aggregation, PAC-1 binding and P-selectin externalization when cells were challenged with thrombin (0.1 U/ml) in presence of DLL-4 compared to samples exposed to thrombin alone (*Figure 4*, A-F). These parameters were considerably attenuated (by 50%, 30.34%, and 23.05%, respectively) upon prior exposure to DAPT (*Figure 4*, A-F). As Notch signaling is propagated through direct cell-cell contact in juxtacrine manner, it is tempting to speculate that cellular proximity achieved within densely packed thrombus milieu would permit interactions between DLL-4 and Notch1 on surfaces of adjacent platelets that would synergize with physiological agonists in realizing thrombus consolidation.

In order to implicate juxtacrine Notch signaling in amplification of platelet activity, we forestalled possible interaction between DLL-4 and Notch1 on adjacent cell surfaces by employing a rabbit polyclonal anti-DLL-4 antibody (20 µg/ml for 5 min) that would block DLL-4. In control samples a non-specific rabbit IgG (20 µg/ml) substituted the antibody against DLL-4. Remarkably, presence of

anti-DLL-4 antibody significantly impaired (by 81.95 %) platelet aggregation induced by thrombin (0.1 U/ml) compared with rabbit IgG-treated counterparts (*Figure 4*, G and H). The extent of drop in aggregation directly correlated with concentration of the blocking antibody in the range from 2 to 20 µg/ml (*Figure 4—figure supplement 1*). Furthermore, shedding of extracellular vesicles from aggregated platelets was also inhibited significantly (by 56.31%) when cells were pre-incubated with anti-DLL-4 antibody compared to rabbit IgG-treated control samples (*Figure 4I*). Above observations were strongly suggestive of juxtracrine Notch signaling operating within the confinement of tightly packed platelet aggregates / thrombi that potentiates platelet stimulation by thrombin.

## Inhibition of γ-secretase attenuates agonist-induced platelet responses

Thrombin and collagen are potent physiological agonists that elicit strong wave of platelet activation through their cognate receptors. Aggregation of washed human platelets induced by diverse agonists (thrombin, 0.25 U/ml; TRAP, 2.5 µM; or collagen, 2.5 µg/ml) were profoundly impaired (by 29.21, 20, and 71.8%, respectively) by DAPT (20 µM) (*Figure 5*, A-F), which, too, retarded TRAP, also known as PAR1-activating peptide (PAR1-AP) and collagen-mediated aggregation (by 46.88 and 28.97%, respectively) in whole blood analyzed from electronic impedance (*Figure 5*, G-J). Thrombin-induced ATP release from platelet dense granules was also attenuated when cells were pre-incubated with DAPT (*Figure 5K*). DBZ, another inhibitor of γ-secretase, also impaired thrombin-induced platelet aggregation (*Figure 5—figure supplement 1*). Interestingly, we also observed significant abrogation of thrombin-induced binding of PAC-1 (*Figure 5—figure supplement 2*, A-D) and fibrinogen (*Figure 5—figure supplement 2*, E and H) to platelet surface integrins, as well as decline in surface externalization of P-selectin (*Figure 5—figure supplement 3*, A-D) and shedding of extracellular vesicles (*Figure 5—figure supplement 4*), when cells were pre-incubated with DAPT (10 µM for 10 min at RT), which was suggestive of critical role of Notch signaling in amplification of agonist-stimulated platelet responses.

Platelet interaction with circulating leukocytes is a sensitive index of state of platelet activity (*Cerletti et al., 2012*; *Ortiz-Muñoz et al., 2014*). In order to implicate Notch signaling in this, platelet-neutrophil and platelet-monocyte aggregates were induced to form in whole blood with addition of TRAP (2 µM, 15 min). Strikingly, percent of cells undergoing aggregation were found to be significantly restrained upon pre-treatment with DAPT (40 µM, 10 min) (*Figure 5*, L-O), which further underlines a role of Notch signaling in platelet-leukocyte interaction and thrombogenesis.

## Inhibition of γ-secretase impairs arterial thrombosis in mice and platelet thrombus generation ex vivo

Platelets play key role in the pathogenesis of arterial thrombosis. In order to implicate Notch signaling in generation of occlusive intramural thrombi in vivo, we studied the effect of pharmacological inhibitor of γ-secretase in a murine model of mesenteric arteriolar thrombosis. Platelets were fluorescently labelled and mice were intraperitoneally administered with either DAPT (50 mg/kg) or vehicle (control). Intramural thrombus was induced by topical application of ferric chloride in exteriorized mesenteric arterioles. Intravital imaging of thrombus was carried out by epifluorescence video microscope equipped with high-speed camera. We observed the time required for first thrombus formation, thrombus growth rate and time to occlusion as indicators to the initiation, propagation and stabilization of thrombus, respectively. Remarkably, mice administered with DAPT (*Video 1*) exhibited significantly delayed thrombus formation compared to vehicle-treated (*Video 2*) animals (mean times to form first thrombus: control, 4.25±1.52 min; DAPT, 7.38±2.94 min ) (*Figure 6A and B*). DAPT also impaired thrombus growth rate compared to vehicle-treated control counterparts (*Figure 6C*) (*Videos 1 and 2*). However, we did not observe significant difference in mean time to stable occlusion (*Figure 6D*). Kaplan-Meier analysis and log-rank test also showed no significant difference in occlusion times between control and DAPT-treated mice (*Figure 6—figure supplement 1*). Above observations attribute a critical role to platelet-specific γ-secretase in initiation and propagation of arterial thrombosis in vivo.

Further, in order explore the role of Notch signaling in generation of thrombus ex vivo, we studied platelet dynamic adhesion and thrombus formation on immobilized collagen under physiological arterial shear (1500 s⁻¹) employing BioFlux microfluidics platform. Washed human platelets were pre-treated with either DAPT (20 µM) or vehicle (control) for 10 min at RT, and allowed to perfuse over the

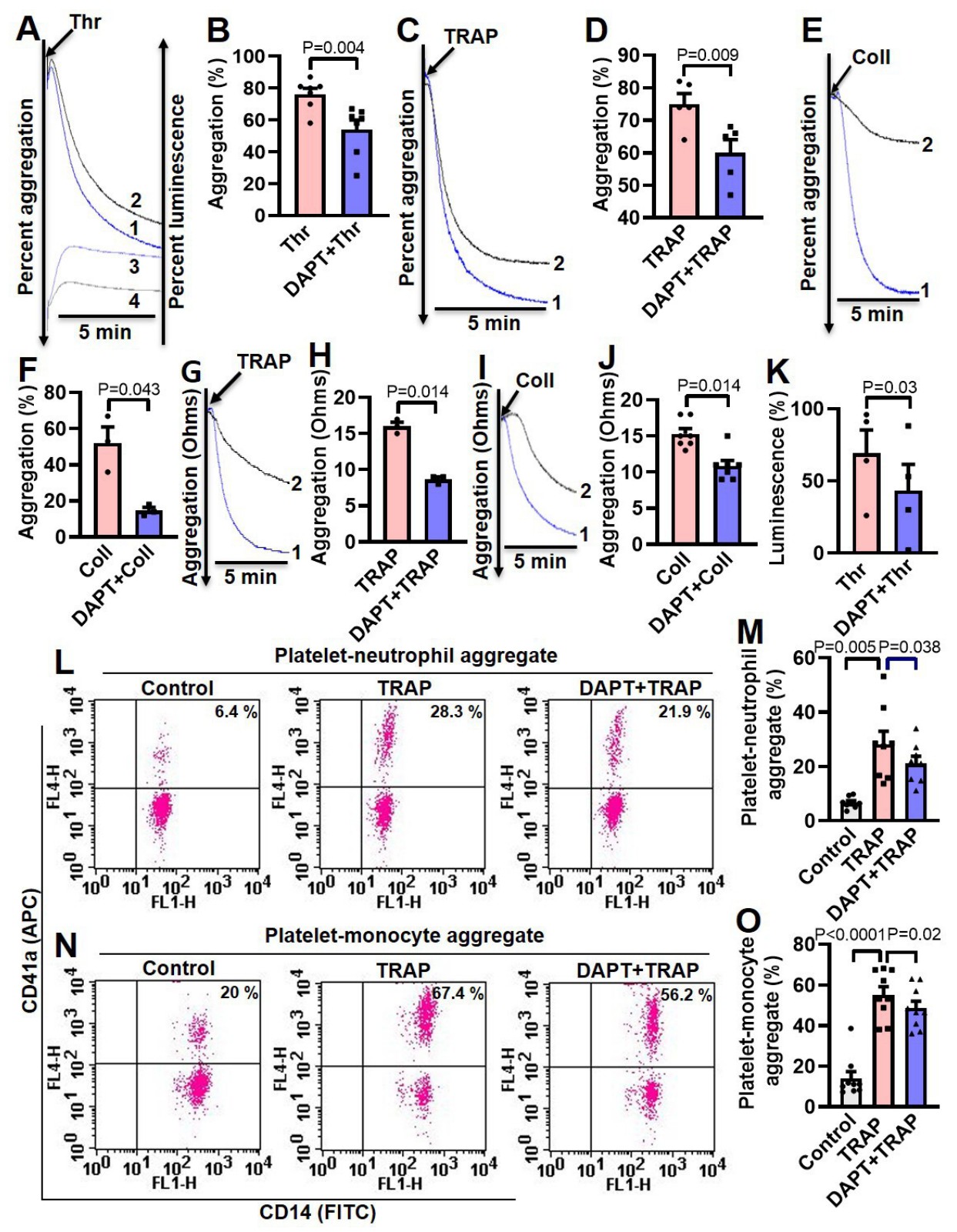

**Figure 5.** Inhibition of γ-secretase attenuates agonist-induced platelet responses. (**A, C** and **E**) Aggregation of washed human platelets induced by thrombin (Thr, 0.25 U/ml), TRAP (2.5 μM), or collagen (Coll, 2.5 μg/ml) in absence (tracing 1) or presence (tracing 2) of DAPT (20 μM) recorded as percent light transmitted. Tracings 3 and 4 in panel A represent secretion of ATP from thrombin-stimulated platelets either in absence or presence of DAPT, respectively. (**G** and **I**) Platelet aggregation in whole blood induced by either TRAP (2 μM) or collagen (2 μg/ml) in absence (tracing 1) or presence

*Figure 5 continued on next page*

*Figure 5 continued*

(tracing 2) of DAPT (40 µM) recorded as change in electrical resistance (impedance). **B** (n=7), **D** (n=5), **F** (n=3), **H** (n=3), and **J** (n=3), corresponding bar chart representing mean platelet aggregation. K, bar diagram representing mean ATP secretion from platelet dense granules (n=4). (**L** and **N**) Flow cytometric analysis of platelet-neutrophil aggregates (**L**) and platelet-monocyte aggregates (**N**) in whole blood stained with anti-CD41a-APC (specific for platelets) and anti-CD14-FITC (specific for neutrophils/monocytes) followed by treatment with TRAP (2 µM) in presence or absence of DAPT (40 µM), as indicated. Amorphous gates were drawn for monocyte (high fluorescence and low SSC) and neutrophil (low fluorescence and high SSC) populations. **M** (n=8) and **O** (n=9), bar diagrams showing percentage of platelet-neutrophil and platelet-monocyte aggregate formation, respectively. Data are representative of at least three different experiments and presented as mean ± SEM. Analyzed by either Student's paired *t*-test (**B, D, F, H, J**, and **K**) or RM one-way ANOVA with Dunnett's multiple comparisons test (**M** and **O**).

The online version of this article includes the following source data and figure supplement(s) for figure 5:

**Source data 1.** Excel sheet shows numerical data of *Figure 5*.

**Figure supplement 1.** Inhibition of γ-secretase attenuates thrombin-induced platelet aggregation.

**Figure supplement 1—source data 1.** Excel sheet shows numerical data of *Figure 5—figure supplement 1*.

**Figure supplement 2.** Inhibition of γ-secretase attenuates thrombin-induced integrin activation.

**Figure supplement 2—source data 1.** Excel sheet shows numerical data of *Figure 5—figure supplement 2*.

**Figure supplement 3.** Inhibition of γ-secretase attenuates thrombin-induced P-selectin externalization.

**Figure supplement 3—source data 1.** Excel sheet shows numerical data of *Figure 5—figure supplement 3*.

**Figure supplement 4.** Inhibition of γ-secretase attenuates thrombin-induced extracellular vesicle release form human platelets.

**Figure supplement 4—source data 1.** Excel sheet shows numerical data of *Figure 5—figure supplement 4*.

collagen-coated surface for 5 min. Interestingly, we observed significant reduction (by 44.1 %) in the total surface area covered by platelet thrombi in the presence of DAPT compared to vehicle-treated control counterparts (*Figure 6*, E and F). This observation also validated a vital role of γ-secretase in thrombosis in ex vivo.

In keeping with above, we next analyzed the contribution of Notch signaling on intrinsic pathway of blood coagulation by employing kaolin-activated thromboelastography. Pre-treatment with DAPT (20 µM) significantly prolonged the reaction time (R) from 5.48±0.52–7.28±1.06 min and attenuated maximum amplitude (MA) by 7.43% (*Figure 6*, G-I; *Supplementary file 1*), which was reflective of delayed formation of thrombus that was significantly less stable as compared with control counterparts with optimal γ-secretase activity. Thus, observations from the in vivo murine model of thrombosis as well as thromboelastography underscored an indispensable role of Notch pathway in determining thrombus stability.

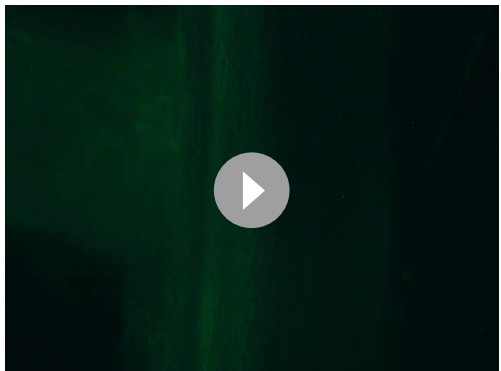

**Video 1.** Ferric chloride-induced mesenteric arteriolar thrombosis in mice pre-administered with DAPT (50 mg/kg). Platelets were fluorescently labelled with DyLight 488 anti-GPIbβ antibody (0.1 µg/g body weight).

https://elifesciences.org/articles/79590/figures#video1

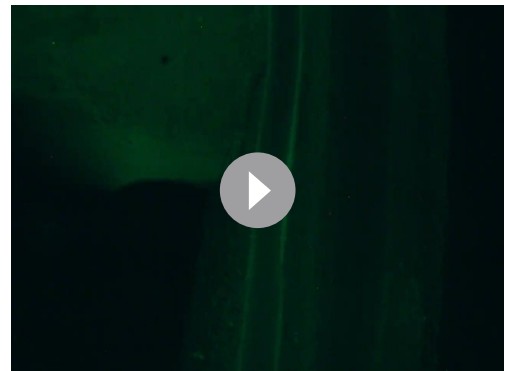

**Video 2.** Ferric chloride-induced mesenteric arteriolar thrombosis in mice pre-administered with vehicle (control). Platelets were fluorescently labelled with DyLight 488 anti-GPIbβ antibody (0.1 µg/g body weight).

https://elifesciences.org/articles/79590/figures#video2

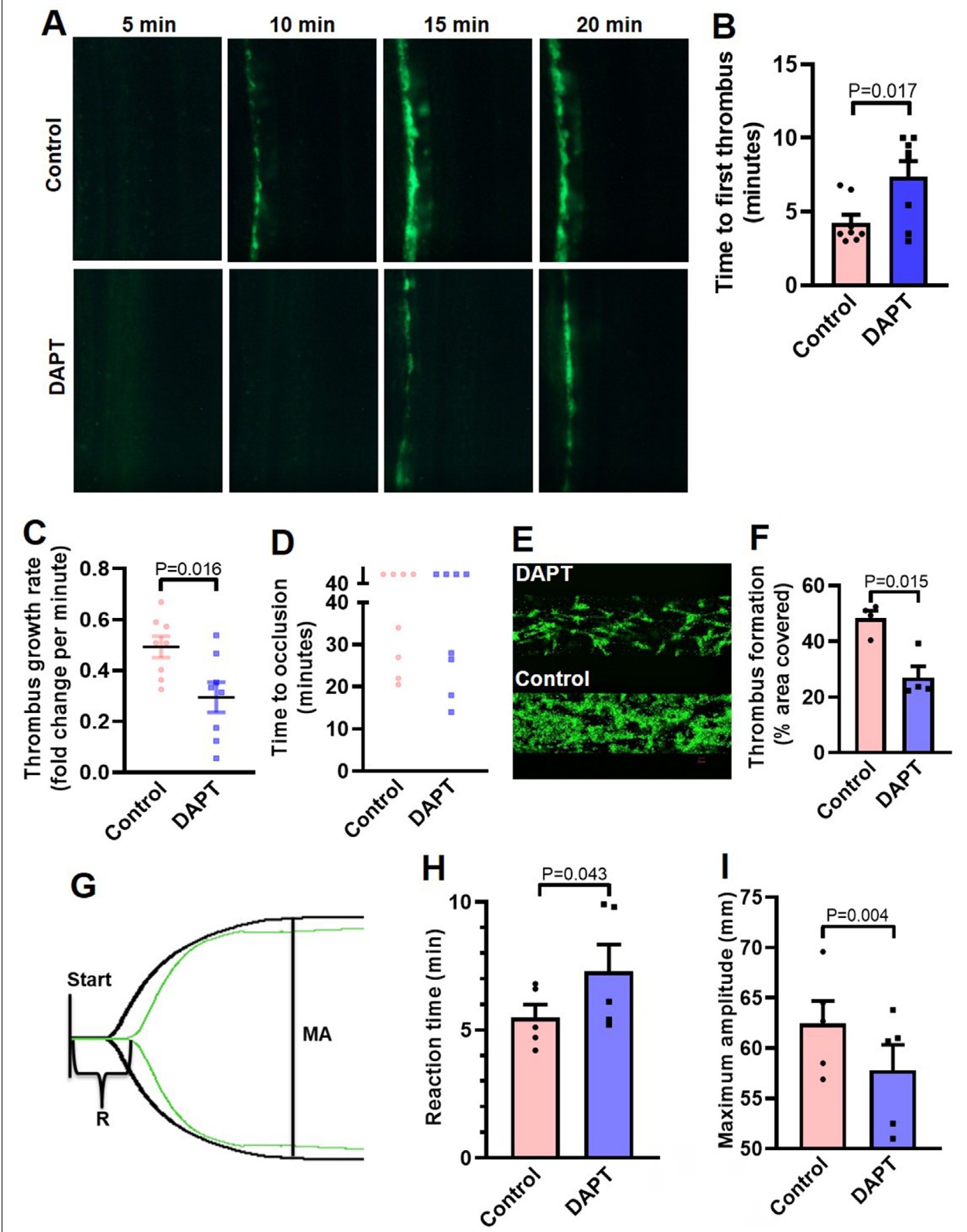

**Figure 6.** Inhibition of γ-secretase precludes arterial thrombosis in mice and platelet thrombus generation in ex vivo. (**A**) Representative time-lapse images showing mesenteric arteriolar thrombosis in mice, pre-administered with either vehicle (control) or DAPT (50 mg/kg) captured 5, 10, 15 or 20 min after ferric chloride-induced injury of the mesenteric arterioles. (**B-D**) Bar diagrams showing time to first thrombus formation (**B**), thrombus growth rate (**C**) and time to occlusion (**D**) (n=8). (**E**) Representative image of platelet accumulation after 5 min of perfusion of human platelets pre-treated with

*Figure 6 continued on next page*

*Figure 6 continued*

either DAPT (20 µM) or vehicle. (**F**) Corresponding bar diagram representing average surface area covered by platelet thrombi after 5 min of perfusion on collagen matrix (n=4). (**G**) Thromboelastogram of kaolin-stimulated citrated whole blood pre-incubated with (green tracing) or without DAPT (black tracing). (**H** and **I**) Bar diagram representing reaction time (R) and maximum amplitude (MA) of the clot, respectively (n=5). Data are representative of at least four individual experiments and presented as mean ± SEM. Analyzed by either unpaired (**B** and **C**) or paired (**F**, **H**, and **I**) Student's *t*-test (unpaired for in vivo and paired for in vitro and ex vivo).

The online version of this article includes the following source data and figure supplement(s) for figure 6:

**Source data 1.** Excel sheet shows numerical data of *Figure 6*.

**Figure supplement 1.** Kaplan-Meier curve representing percent of occluded mesenteric arteries at varying time points in mice pre-administered with either vehicle (control) or DAPT (50 mg/kg), as indicated (n=8).

**Figure supplement 1—source data 1.** Excel sheet shows numerical data of *Figure 6—figure supplement 1*.

## Discussion

The Notch signaling has been implicated in production of megakaryocytes and platelets from CD34[+] cells (*Poirault-Chassac et al., 2010*). However, expression of Notch receptors and its functionality in human platelets has remained unexplored. Notch signaling is mediated through four isoforms of mammalian Notch receptors, namely Notch1 to Notch4, which interact with five independent Notch ligands, DLL-1,–3, –4 and Jag-1 and –2 (*Kopan and Ilagan, 2009*). In this report we have demonstrated that, enucleate platelets have notable expression of Notch1 and its ligand DLL-4, which function in a non-canonical manner to synergize with physiological platelet agonists, leading to generation of prothrombotic phenotype. Although platelets have limited protein-synthesizing ability, exposure to thrombin instigated significant translation of DLL-4 and Notch1 in puromycin-sensitive manner, thus adding them to the growing repertoire of platelet translatome. Strikingly, thrombin, too, provoked translocation of these peptides to platelet surface membrane, raising possibility of juxtacrine DLL-4-Notch1 interaction within the confinement of platelet aggregates. DLL-4 stimulated significant rise in expression of NICD, the cleavage product of Notch, in a γ-secretase-dependent manner, that signifies the existence of functional DLL-4-Notch1-NICD signaling axis in platelets. This is not out of place here to mention that γ-secretase, too, is responsible for cleavage of amyloid-precursor proteins (APP) releasing amyloid-β (Aβ) (*Tarassishin et al., 2004*). Platelets, which contribute to 95% of circulating APP in body (*Li et al., 1999*; *Davies et al., 1997*; *Bush et al., 1990*), are known to generate Aβ$_{40}$ upon stimulation with physiological agonists like thrombin or collagen in a PKC-dependent manner (*Smith and Broze, 1992*; *Skovronsky et al., 2001*), leading to rise in local concentration of Aβ within the thrombus (*Skovronsky et al., 2001*; *Smith, 1997*).

Platelets are central players in hemostasis and pathological thrombosis that can lead to occlusive cardiovascular pathologies like myocardial infarction and ischemic stroke. Upon activation platelet surface integrins α$_{IIb}$β$_3$ switch to an open conformation, which allows high-affinity binding of fibrinogen and cell-cell aggregate formation, surface mobilization of P-selectin and rise in intracellular free calcium. In our quest to explore the non-genomic role of Notch pathway in platelet biology, we discovered that DLL-4 and not DLL-1 was able to instigate significant binding of PAC-1 (that recognizes the open conformation of α$_{IIb}$β$_3$) and fibrinogen to platelets, associated with exocytosis of contents of alpha and dense granules, which was consistent with switch to a 'pro-active / pro-thrombotic' phenotype. DLL-4, too, provoked Ca$^{2+}$ influx leading to substantial rise in intracellular Ca$^{2+}$, extracellular vesicle shedding, platelet-leukocyteaggregate formation and increase in platelet tyrosine phosphoproteome, all hallmarks of stimulated platelets. Notably, inhibition of γ-secretase employing two pharmacologically different compounds significantly impaired DLL-4-mediated 'pro-activating' effects on platelets. Inhibitors of either PI3K or protein kinase C evoked remarkable decrease in PAC1 binding, which underlines contributions of these enzymes in DLL-4-induced integrin activation. The phosphorylations of PI3K and AKT (downstream of PI3K) in platelets were significantly boosted in presence of DLL-4, which were attenuated upon pre-treatment with DAPT. In sum, above findings are strongly suggestive of non-canonical signaling triggered by DLL-4 in human platelets that operates in γ-secretase-PI3K-AKT-dependent manner leading to platelet activation.

Thrombin is a potent physiological agonist that induces platelet aggregation and secretion through cognate PAR receptors. As thrombin amplifies expression of DLL-4 on platelet surface and DLL-4, in turn, induces platelet activation signaling, we asked whether DLL-4 synergizes with thrombin in

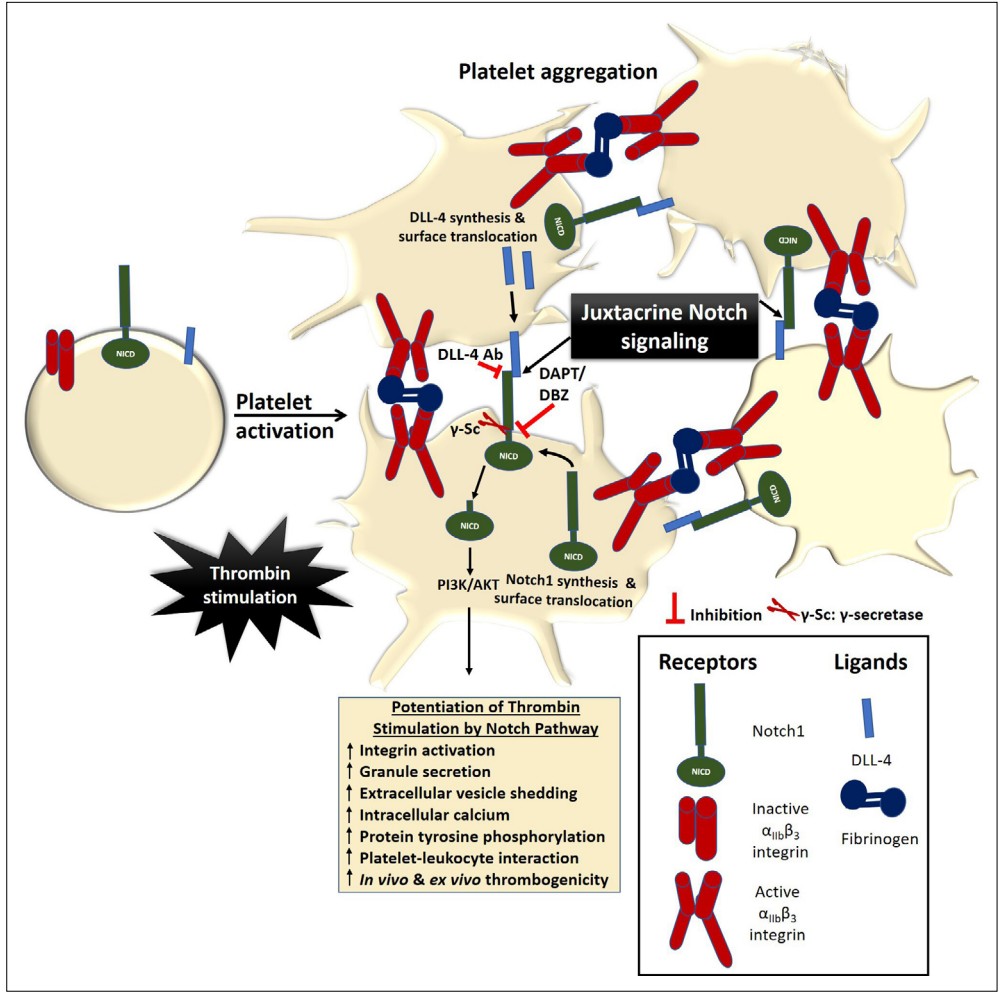

**Figure 7.** Scheme depicting the role of Notch signaling in potentiating agonist-induced platelet stimulation. Juxtacrine interaction between DLL-4 and Notch1 expressed on surfaces of agonist-stimulated platelets that remain in close proximity within platelet aggregates, leading to potentiation of thrombin signaling and consolidation of thrombus. The juxtacrine responses are blocked by employing either anti-DLL-4 antibody (blocking antibody) or inhibitors of γ-secretase.

stimulation of platelets. Pre-treatment of cells with DLL-4 followed by low dose (0.1 U/ml) thrombin significantly upregulated platelet aggregation, PAC-1 binding and P-selectin exposure elicited by thrombin alone, which underlines a potentiating effect by DLL-4. These responses were considerably abrogated by inhibitor of γ-secretase that further authenticates contribution of Notch signaling in platelet activation. Platelet aggregation induced by diverse agonists including thrombin, collagen and TRAP was also restrained by inhibiting γ-secretase activity. Remarkably, pharmacological inhibition of γ-secretase significantly impaired thrombus formation in a murine model of mesenteric arteriolar thrombosis and platelet thrombus generation ex vivo. As our laboratory and others have demonstrated Aβ to be a potent stimulus for platelets with thrombogenic attributes (*Sonkar et al., 2014*; *Shen et al., 2008*; *Canobbio et al., 2014*), pharmacologic inhibition of γ-secretase in platelets would prohibit release of both NICD and Aβ that may be envisaged as an effective multimodal anti-thrombotic strategy leading to thrombus destabilization. Kaolin-activated thromboelastography, too, validated delayed formation of thrombus that was significantly less stable compared with control counterpart having optimal γ-secretase activity. Taken together, above observations underscored seminal contribution from DLL-4-Notch1-γ-secretase axis in amplification of agonist-mediated platelet responses and determination of thrombus stability.

As direct cell-cell contact is the mainstay of Notch signaling, it is reasonable to speculate interactions between DLL-4 and Notch1 on surfaces of adjacent platelets, which are closely approximated

within the densely packed thrombus milieu. In order to validate it, we blocked proximity between DLL-4 and Notch1 by pre-incubating platelets with a rabbit polyclonal anti-DLL-4 antibody, followed by stimulation with thrombin. In control samples, a non-specific rabbit IgG substituted the antibody against DLL-4. Remarkably, presence of blocking antibody significantly impaired platelet aggregation evoked by thrombin compared with the rabbit IgG-treated counterparts. The extent of drop in aggregation directly correlated with concentration of the blocking antibody. In keeping with it, shedding of extracellular vesicles from aggregated platelets was also potentially inhibited when cells were pre-incubated with anti-DLL-4 antibody, and not with rabbit IgG. Above observations were overwhelmingly supportive of juxtacrine Notch signaling operating within the tightly packed platelet aggregate/thrombus milieu that potentiates platelet stimulation by physiological agonists.

In conclusion, we provide compelling evidence in favour of a PI3K /AKT-dependent non-canonical Notch signaling pathway operative in enucleate platelets that contributes significantly to the stability of occlusive arterial thrombus as well as to platelet activation instigated by thrombin through juxtacrine interactions (*Figure 7*). We demonstrate that inhibitors of γ-secretase, which downregulate Notch signaling in platelets, could be effective anti-platelet agents. Besides, antibody against DLL-4 may be employed therapeutically to forestall DLL-4-Notch1 interaction on surfaces of adjacent platelets as a potential anti-thrombotic approach.

## Acknowledgements

This research was supported by J C Bose National Fellowship (JCB/2017/000029) and grants received by D Dash from the Indian Council of Medical Research (ICMR) under CAR (71/4/2018-BMS/CAR), Department of Biotechnology (DBT) (BT/PR-20645/BRB/10/1541/2016) and Science and Engineering Research Board (SERB) (EMR/2015/000583), Government of India. SN Chaurasia, M Ekhlak and V Singh are recipients of ICMR-Scientist-C, CSIR-SRF and UGC-SRF support, respectively. D Dash acknowledges assistance from the Humboldt Foundation, Germany. The funders had no role in study design, data collection and interpretation, or decision to submit the work for publication.

## Additional information

### Funding

| Funder | Grant reference number | Author |
|---|---|---|
| JC Bose National Fellowship | JCB/2017/000029 | Debabrata Dash |
| Indian Council of Medical Research | 71/4/2018-BMS/CAR | Debabrata Dash |
| Department of Biotechnology | BT/PR-20645/BRB/10/1541/2016 | Debabrata Dash |
| Science and Engineering Research Board | EMR/2015/000583 | Debabrata Dash |
| Indian Council of Medical Research | | Susheel N Chaurasia |
| Council of Scientific and Industrial Research, India | | Mohammad Ekhlak |
| University Grants Commission India | | Vipin Singh |
| Humboldt Foundation | | Debabrata Dash |

The funders had no role in study design, data collection and interpretation, or the decision to submit the work for publication.

### Author contributions

Susheel N Chaurasia, Conceptualization, Formal analysis, Validation, Investigation, Visualization, Methodology, Writing - original draft, Writing - review and editing; Mohammad Ekhlak, Geeta Kushwaha,

Vipin Singh, Formal analysis, Investigation, Methodology; Ram L Mallick, Formal analysis, Investigation, Methodology, Wrote the preliminary draft of the PCR data; Debabrata Dash, Conceptualization, Supervision, Funding acquisition, Validation, Investigation, Visualization, Writing - original draft, Project administration, Writing - review and editing

### Author ORCIDs
Susheel N Chaurasia (ID) http://orcid.org/0000-0002-4207-8805
Mohammad Ekhlak (ID) http://orcid.org/0000-0003-1041-2175
Geeta Kushwaha (ID) http://orcid.org/0000-0002-3470-8635
Vipin Singh (ID) http://orcid.org/0000-0001-7078-4320
Ram L Mallick (ID) http://orcid.org/0000-0002-2407-5686
Debabrata Dash (ID) http://orcid.org/0000-0001-7291-2453

### Ethics
Human subjects: Blood samples were drawn from healthy adult human participants after obtaining written informed consent, strictly as per recommendations and as approved by the Institutional Ethical Committee of the Institute of Medical Sciences, Banaras Hindu University (Approval No. Dean/2015-16/EC/76).

The animal study was ethically approved by the Central Animal Ethical Committee of Institute of Medical Sciences, Banaras Hindu University (Approval No. Dean/2017/CAEC/83). All efforts were made to minimize the number of animals used, and their suffering.

### Decision letter and Author response
Decision letter https://doi.org/10.7554/eLife.79590.sa1
Author response https://doi.org/10.7554/eLife.79590.sa2

## Additional files

### Supplementary files
• Supplementary file 1. Thromboelastogram of kaolin-stimulated citrated whole blood pre-treated with or without DAPT. Data are representative of five individual experiments and presented as mean ± SEM, analyzed by Student's paired *t*-test.

• MDAR checklist

### Data availability
All the data are available in the main text or the supporting information. Source data files have been provided for each figure included either in the manuscript or supplemental data.

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
