## [Editor Report]

Advances in the discovery of novel anti-platelet therapeutics remains an unmet need. This manuscript by Chaurasia et al., describes a novel signaling pathway involving Notch1 and its ligand, Δ-like ligand-4 (DLL4) in driving platelet activation and thrombus formation. The authors provide convincing mechanistic studies to show that blockade of this pathway may serve as a new therapeutic approach to prevent/treat thrombosis. The work will be of great interest to individuals in the hematology and thrombosis field.

---

## [Decision Letter]

**Decision letter after peer review:**

Thank you for submitting your article "Notch signaling functions in non-canonical juxtacrine manner in platelets to amplify thrombogenicity" for consideration by *eLife*. Your article has been reviewed by 3 peer reviewers, and the evaluation has been overseen by a Reviewing Editor and Mone Zaidi as the Senior Editor. The reviewers have opted to remain anonymous.

Essential revisions:

Each of the three reviewers commented on the experimental approach being rigorous, data clearly presented, and claims/conclusions well supported by the data. The reviewers have provided some suggestions to help improve upon some aspects of the paper. Detailed comments are included below.

*Reviewer #1 (Recommendations for the authors):*

This manuscript describes a new signaling pathway of platelet activation. The manuscript may be considered for publication following the authors reply to the comments below.

Authors may consider the following comments:

1. English editing for whole manuscript.

2. May fix the conclusion part. Such as relocating the part describing "involvement of γ-secretase in the cleavage of cleavage of amyloid-precursor proteins (APP) releasing amyloid-β (Aβ), etc." to other part of the discussion. May focus on describing the involvement of DLL-4-Notch1 axis.

3. In Figure 3I, What are the tyrosine phosphorylated proteins? The name of specific proteins may be mentioned against specific kDa. A clear representative blot may be used.

4. In Figure 3J, along with pAkt, the pPI3K blot may also be used.

5. In Figure 4A and G, the percentage aggregation may mentioned in the Y axis. In 4A, only thrombin treatment shows 40% aggregation, where as in Figure 4G, it is 60% aggregation.

6. In Figure 5A,C,E,G,I, the Y axis may represent the percentage aggregation.

*Reviewer #2 (Recommendations for the authors):*

1. Figure 4G: Tracing 1 should be explained in the figure legend.

2. Aggregation of washed human platelets induced by thrombin in presence of DBZ should be presented as supplementary data.

3. Page 6: "Figure 1—figure supplement 1C and S2C" should be replaced with "Figure 1—figure supplement 1C and figure supplement 2C".

*Reviewer #3 (Recommendations for the authors):*

1. The authors should show the full blot with molecular weight markers in the supplemental data to demonstrate the specificity of the antibodies.

2. In several panels, such as Figure 1F, the value of each group has a pretty big range: Resting platelet AU is from 0.2 – 0.8; thrombin stimulated platelet AU is from 0.2 – 1.0. Is there any correlation among 3 groups? For example, the 0.8 dot in resting platelets is the 1.0 dot in thrombin stimulated platelets.

3. For this study, the author used 15 μg/mL DLL-4 to stimulate platelets. The author should give some justification for this dose: (e.g., is it close to the physiological concentration of DLL-4?).

4. The logic flow on how NICD was identified as a substrate for calpeptin is unclear. Did the authors do a series of experiments to narrow down to calpeptin? Can the authors provide justification for this line of reasoning?

5. The TRAP is the PAR1 activation peptide. Since platelets express both PAR1 and PAR4 as thrombin receptors and both are activated by thrombin, the authors should refer to TRAP as PAR1-AP and give some justification for not using PAR4-activation peptide in the experiment design as well.

---

## [Author Response]

Reviewer #1 (Recommendations for the authors):This manuscript describes a new signaling pathway of platelet activation. The manuscript may be considered for publication following the authors reply to the comments below.Authors may consider the following comments:1. English editing for whole manuscript.

We thank the Reviewer for her/his observation. We have now thoroughly rechecked and reviewed the entire manuscript for grammatical and syntactical errors and edited as required.

2. May fix the conclusion part. Such as relocating the part describing "involvement of γ-secretase in the cleavage of cleavage of amyloid-precursor proteins (APP) releasing amyloid-β (Aβ), etc." to other part of the discussion. May focus on describing the involvement of DLL-4-Notch1 axis.

We thank the Reviewer for the valuable note. Accordingly, we have now relocated the concerned statement in Conclusion part in our revised manuscript as suggested.

3. In Figure 3I, What are the tyrosine phosphorylated proteins? The name of specific proteins may be mentioned against specific kDa. A clear representative blot may be used.

Identification and analysis of tyrosine phosphoproteome would need MS-based approach, which is beyond the scope of this study. We have now replaced the blot as suggested (Figure 3I).

4. In Figure 3J, along with pAkt, the pPI3K blot may also be used.

We thank the Reviewer for this perspective to improve the manuscript. Accordingly, we have now carried out additional experiments to determine the effect of DLL-4 on phosphorylation of p85 regulatory subunit of PI3K, either in absence or presence of DAPT, in platelets. The data have been incorporated in the revised manuscript (Figure 3, J and K).

5. In Figure 4A and G, the percentage aggregation may mentioned in the Y axis. In 4A, only thrombin treatment shows 40% aggregation, where as in Figure 4G, it is 60% aggregation.

As suggested by the Reviewer, we have now mentioned the percent aggregation against Y-axis in the concerned Figures. Variation in percent aggregation was expected as the samples were collected from different individuals.

6. In Figure 5A,C,E,G,I, the Y axis may represent the percentage aggregation.

We have made changes in our revised manuscript in line with Reviewer’s suggestion.

Reviewer #2 (Recommendations for the authors):1. Figure 4G: Tracing 1 should be explained in the figure legend.

As suggested by the Reviewer, we have now explained tracing 1 in the figure legend.

2. Aggregation of washed human platelets induced by thrombin in presence of DBZ should be presented as supplementary data.

We thank the Reviewer for the suggestion. Accordingly, we have now carried out additional experiments where platelet aggregation has been induced with thrombin either in absence or presence of DBZ. The data have been incorporated in the Supplementary Section (Figure 5—figure supplement 1) as suggested.

3. Page 6: "Figure 1—figure supplement 1C and S2C" should be replaced with "Figure 1—figure supplement 1C and figure supplement 2C".

As suggested by the Reviewer, we have now replaced “Figure 1—figure supplement 1C and S2C” with “Figure 1—figure supplement 1C and figure supplement 2C” in the revised manuscript.

Reviewer #3 (Recommendations for the authors):1. The authors should show the full blot with molecular weight markers in the supplemental data to demonstrate the specificity of the antibodies.

As suggested by the Reviewer, we have now shown molecular weight markers on respective full blots. It may be stated that, we have already submitted all the Western blots as source data files.

2. In several panels, such as Figure 1F, the value of each group has a pretty big range: Resting platelet AU is from 0.2 – 0.8; thrombin stimulated platelet AU is from 0.2 – 1.0. Is there any correlation among 3 groups? For example, the 0.8 dot in resting platelets is the 1.0 dot in thrombin stimulated platelets.

We thank the reviewer for her/his observation and concern. We would like to submit that, Arbitrary Unit (AU) 0.8 for resting platelets, 1.0 for thrombin-stimulated platelets, and 0.7 for puromycin-pre-treated thrombin-stimulated platelets represent data obtained from the same donor. For clarification, kindly refer to the raw data pertaining to Figure 1F submitted as source data file.

3. For this study, the author used 15 μg/mL DLL-4 to stimulate platelets. The author should give some justification for this dose: (e.g., is it close to the physiological concentration of DLL-4?).

Earlier studies have employed DLL-4 at concentrations ranging from 1 to 20 µg/ml (Michaels et al., 2021, Hu et al., 2011). In our report, we have observed 15 µg/ml to be the optimal concentration of DLL-4 to elicit signaling in platelets. The dissociation constant (K_d_) for DLL-4 and Notch1 binding ranges from 7.5 µM to 12.7 µM (Luca et al., 2015). Thus, the concentration of DLL-4 employed in our study (~200 nM) is most likely to be physiologically relevant.

4. The logic flow on how NICD was identified as a substrate for calpeptin is unclear. Did the authors do a series of experiments to narrow down to calpeptin? Can the authors provide justification for this line of reasoning?

Calpain is a ca^2+^-dependent thiol protease that is known to be activated in platelets stored at 37 °C and following stimulation with ionophore A23187 (Wadhawan et al., 2004). Several signaling and structural proteins including FAK (focal adhesion kinase), *Src*, Btk (Bruton’s tyrosine kinase), protein tyrosine phosphatase 1B (PTP1B), ABP (actin-binding protein), filamin, talin, myosin (heavy chain) and spectrin are well known substrates of calpain in human platelets (Fox et al., 1985, Fox et al., 1987, Fox et al., 1990, Fox et al., 1991, Mukhopadhyay et al., 2001, Wadhawan et al., 2004). Calpain has been projected as a regulator of platelet outside-in signaling, aggregation and clot retraction (Croce et al., 1999, Schoenwaelder et al., 1997, Kuchay and Chishti, 2007). This prompted us to ask whether NICD is also a substrate of calpain in platelets under storage or upon ionophore A23187-stimulation. Interestingly, we observed remarkable drop in the level of NICD when platelets were either stored at 37 °C ͦbut not at 22 °C ͦor stimulated with ionophore A23187. This decrease was reverted when cells were pre-incubated with calpain inhibitors like calpeptin or ALLN, and calcium chelators like EGTA. These observations add NICD to the repertoire of calpain substrates in human platelets.

5. The TRAP is the PAR1 activation peptide. Since platelets express both PAR1 and PAR4 as thrombin receptors and both are activated by thrombin, the authors should refer to TRAP as PAR1-AP and give some justification for not using PAR4-activation peptide in the experiment design as well.

We thank the reviewer for constructive suggestion. Accordingly, we have now referred TRAP as PAR1-activating peptide (PAR1-AP) in our revised manuscript. Platelets are known to express both PAR1 and PAR4 as dual receptor system for thrombin-mediated signaling (Kahn et al., 1999, Coughlin, 2000). However, keeping with varying affinity of receptors towards ligand, PAR1 needs low-dose thrombin for activation of platelets whereas PAR4 requires high thrombin concentration for signal transmission (Duvernay et al., 2017). Hence, TRAP has been employed as an activation peptide for PAR1 as most of the experiments in our study were carried-out with low-dose of thrombin.

References:

Coughlin, S. R. 2000. Thrombin signalling and protease-activated receptors. *Nature,* 407**,** 258-64.

Croce, K., Flaumenhaft, R., Rivers, M., Furie, B., Furie, B. C., Herman, I. M. And Potter, D. A. 1999. Inhibition of calpain blocks platelet secretion, aggregation, and spreading. *J Biol Chem,* 274**,** 36321-7.

Duvernay, M. T., Temple, K. J., Maeng, J. G., Blobaum, A. L., Stauffer, S. R., Lindsley, C. W. And Hamm, H. E. 2017. Contributions of Protease-Activated Receptors PAR1 and PAR4 to Thrombin-Induced GPIIbIIIa Activation in Human Platelets. *Mol Pharmacol,* 91**,** 39-47.

Fox, J. E., Austin, C. D., Reynolds, C. C. And Steffen, P. K. 1991. Evidence that agonist-induced activation of calpain causes the shedding of procoagulant-containing microvesicles from the membrane of aggregating platelets. *J Biol Chem,* 266**,** 13289-95.

Fox, J. E., Goll, D. E., Reynolds, C. C. And Phillips, D. R. 1985. Identification of two proteins (actin-binding protein and P235) that are hydrolyzed by endogenous ca^2+^-dependent protease during platelet aggregation. *J Biol Chem,* 260**,** 1060-6.

Fox, J. E., Reynolds, C. C. And Austin, C. D. 1990. The role of calpain in stimulus-response coupling: evidence that calpain mediates agonist-induced expression of procoagulant activity in platelets. *Blood,* 76**,** 2510-9.

Fox, J. E., Reynolds, C. C., Morrow, J. S. And Phillips, D. R. 1987. Spectrin is associated with membrane-bound actin filaments in platelets and is hydrolyzed by the ca^2+^-dependent protease during platelet activation. *Blood,* 69**,** 537-45.

Hu, W., Lu, C., Dong, H. H., Huang, J., Shen, D. Y., Stone, R. L., Nick, A. M., Shahzad, M. M., Mora, E., Jennings, N. B., Lee, S. J., Roh, J. W., Matsuo, K., Nishimura, M., Goodman, B. W., Jaffe, R. B., Langley, R. R., Deavers, M. T., Lopez-Berestein, G., Coleman, R. L. And Sood, A. K. 2011. Biological roles of the Δ family Notch ligand Dll4 in tumor and endothelial cells in ovarian cancer. *Cancer Res,* 71**,** 6030-9.

Kahn, M. L., Nakanishi-Matsui, M., Shapiro, M. J., Ishihara, H. And Coughlin, S. R. 1999. Protease-activated receptors 1 and 4 mediate activation of human platelets by thrombin. *J Clin Invest,* 103**,** 879-87.

Kuchay, S. M. And Chishti, A. H. 2007. Calpain-mediated regulation of platelet signaling pathways. *Curr Opin Hematol,* 14**,** 249-54.

Luca, V. C., Jude, K. M., Pierce, N. W., Nachury, M. V., Fischer, S. And Garcia, K. C. 2015. Structural biology. Structural basis for Notch1 engagement of Δ-like 4. *Science,* 347**,** 847-53.

Michaels, Y. S., Edgar, J. M., Major, M. C., Castle, E. L., Zimmerman, C., Yin, T., Hagner, A., Lau, C., Ibañez-Rios, M. I., Knapp, D. J. H. F. And Zandstra, P. W. 2021. DLL4 and VCAM1 enhance the emergence of T cell-competent hematopoietic progenitors from human pluripotent stem cells. *bioRxiv***,** 2021.11.26.470145.

Mukhopadhyay, S., Ramars, A. S., Ochs, H. D. And Dash, D. 2001. Bruton's tyrosine kinase is a substrate of calpain in human platelets. *FEBS Lett,* 505**,** 37-41.

Schoenwaelder, S. M., Yuan, Y., Cooray, P., Salem, H. H. And Jackson, S. P. 1997. Calpain cleavage of focal adhesion proteins regulates the cytoskeletal attachment of integrin alphaIIbbeta3 (platelet glycoprotein IIb/IIIa) and the cellular retraction of fibrin clots. *J Biol Chem,* 272**,** 1694-702.

Wadhawan, V., Karim, Z. A., Mukhopadhyay, S., Gupta, R., Dikshit, M. And Dash, D. 2004. Platelet storage under in vitro condition is associated with calcium-dependent apoptosis-like lesions and novel reorganization in platelet cytoskeleton. *Arch Biochem Biophys,* 422**,** 183-90.